



# 1 Fatty acid carbon isotopes: a new indicator of marine
# 2 Antarctic paleoproductivity?

Kate Ashley[1], James Bendle[1], Xavier Crosta[2], Johan Etourneau[2,3], Philippine Campagne[2,4],
Harry Gilchrist[1], Uthmaan Ibraheem[1], Sarah Greene[1], Sabine Schmidt[2], Yvette Eley[1],
Guillaume Massé[4,5]
[1]School of Geography, Earth and Environmental Sciences, University of Birmingham, Edgbaston, Birmingham,
B15 2TT, UK
[2]EPOC, UMR-CNRS 5805, Université de Bordeaux, 33615 Pessac, France
[3]EPHE/PSL Research University, 75014 Paris, France
[4]LOCEAN, UMR CNRS/UPCM/IRD/MNHN 7159, Université Pierre et Marie Curie, 4 Place Jussieu, 75252
Paris, France
[5]TAKUVIK, UMI 3376 UL/CNRS, Université Laval, 1045 avenue de la Médecine, Quebec City, Quebec,
Canada G1V 0A6
*Correspondence to:* James Bendle (j.bendle@bham.ac.uk)
**Abstract**
The Antarctic coastal zone is an area of high primary productivity, particularly within coastal polynyas where
large phytoplankton blooms and drawdown of $CO_2$ occur. Reconstruction of historical primary productivity
changes, and the associated driving factors, could provide baseline insights on the role of these areas as sinks for
atmospheric $CO_2$, especially in the context of projected changes in coastal Antarctic sea ice. Here we investigate
the potential for using carbon isotopes ($\delta^{13}C$) of fatty acids in marine sediments as a proxy for primary
productivity. We use a highly resolved sediment core from off the coast of Adélie Land spanning the last ~400
years and monitor changes in the concentrations and $\delta^{13}C$ of fatty acids along with other proxy data from the
same core. We discuss the different possible drivers of their variability and argue that $C_{24}$ fatty acid $\delta^{13}C$
predominantly reflects phytoplankton productivity in open water environments, while $C_{18}$ fatty acid $\delta^{13}C$
reflects productivity in the marginal ice zone. These new proxies have implications for better understanding
carbon cycle dynamics in the Antarctica coastal zone in future paleoclimate studies.

**1 Introduction**
Antarctic coastal zones are important players in the global carbon cycle. The deep ocean is ventilated in these
regions as part of the Southern Ocean overturning circulation, allowing waters rich in nutrients and $CO_2$ to be
upwelled to the surface. In the absence of biological activity, most of the $CO_2$ would be leaked to the
atmosphere. However, coastal polynyas within the Antarctic margin are areas of very high primary productivity
during the spring and summer months (e.g. Arrigo et al., 2008) that rapidly reduces $CO_2$ to low levels through
photosynthesis (Arrigo and van Dijken, 2003; Arrigo et al., 2008), resulting in surface water $CO_2$
undersaturation with respect to atmospheric $CO_2$ (Tortell et al., 2011). The subsequent export and burial of the
organic carbon produced during these intense phytoplankton blooms can significantly lower atmospheric $CO_2$
concentrations (Sigman and Boyle, 2000). Therefore, any change in the consumption of these nutrients by



phytoplankton, or any change in phytoplankton community structure, may affect the air-sea $CO_2$ exchange in
this region.
Records of past phytoplankton productivity offer an opportunity to document the drivers of primary productivity
at different timescales from pluri-decadal to millennial. In the Antarctic coastal zone past work has focused on
records of organic carbon, biogenic silica and diatom abundances (Leccaroni et al., 1998; Frignani et al., 1998;
Denis et al., 2009; Peck et al., 2015). These proxies however may provide a biased view of phytoplankton
productivity as they only record a signal of siliceous productivity and may suffer from alteration during settling
and burial (Beucher et al., 2004; Tréguer et al., 2017). As such, there is no robust understanding of how such
records respond to surface water $CO_2$ which is of major importance in the context of Antarctic coastal sea ice
changes.
Here we investigate the use of compound specific carbon isotope analysis ($\delta^{13}C$) of algal fatty acids (FAs) in
marine sediments as a potential integrative proxy for reconstructing primary productivity in a polynya
environment. We use samples from core DTGC2011, a 4.69 m sediment core recovered from offshore Adélie
Land, East Antarctica, spanning the last ~400 years. The core chronology is based on radiocarbon dates and
confirmed by $^{210}Pb$ excess activity measurements, which indicate that DTGC2011 spans the 1580-2000 C.E.
period with a mean sedimentation rate of ~1 cm $yr^{-1}$ (Supplementary Information S1). In order to understand the
signal recorded by the FAs, we estimate the most likely biological source of these compounds and the habitat
and season of production. Moreover, we compare downcore changes in FA concentrations and $\delta^{13}C$ with other
proxy data from the same core.

**Environmental setting**
The Adélie drift is located in the Dumont D'Urville Trough in the Adélie Basin, ca. 35 km offshore from Adélie
Land (Fig. 1). This is a 1000 m deep, glacially scoured depression on the East Antarctic continental shelf,
bounded to the east by the Adélie Bank. Sea ice plays a key role on the dynamics of the region, with both fast
ice and pack ice present off the coast of Adélie Land. A large bank of fast ice forms annually between 135 and
142°E, and extends up to 120 km away from the coast (Massom et al., 2009). On the north edge of this fast ice
buttress is an inlet of open water forming a polynya, an area of open water surrounded by sea ice (Bindoff et al.,

67    2000).

The Adélie Coast is characterized by extremely high primary productivity, with phytoplankton assemblages
dominated by diatoms (Beans et al., 2008). The site itself is located close to the Dumont D'Urville polynya
(DDUP), but is also directly downwind and downcurrent of the much larger and highly productive Mertz
Glacier polynya (MGP) to the east (Arrigo and van Dijken, 2003).
The region is affected by various water masses. High Salinity Shelf Water (HSSW) is formed on the shelf in
coastal polynyas as a result of sea ice production and the associated brine rejection. HSSW flows out of the shelf
through the Adélie sill at 143°E (Fig. 1). Modified Circumpolar Deep Water (mCDW) is a warm, macronutrient-
rich and salty water mass which upwells onto the continental shelf through channels in the shelf break. mCDW
has been observed to upwell across the shelf break near the Mertz Glacier at 144°E (Williams et al., 2008) (Fig.



1). The Antarctic Coastal Current, also known as the East Wind Drift, flows westward often adjacent to ice
shelves (Thompson et al., 2018). The Antarctic Surface Water (AASW) is a widespread water mass which
extends across the continental shelf and has a surface mixed layer varying from a shallow (ca. 10 m), warmer
and fresher layer in summer to a deeper (ca. 100 m), colder layer in winter. This is also transported westward
along with the Antarctic Coastal Current (Martin et al., 2017). Surface waters along the Adélie coast have
relatively high concentrations of nitrate, silica and phosphorus, with spatially variable levels of Fe which may be
due to re-suspension of sediments and calving of ice (Vaillancourt et al., 2003; Sambrotto et al., 2003).
**2 Materials and Methods**
*Fatty acids*
One hundred and thirty-five sediment samples were taken for organic geochemical analyses, sampled at 1 cm
intervals in the top 50 cm, 2 cm intervals between 50 and 100 cm, and 5 cm intervals until 458 cm. Lipid
extractions were completed at the University of Birmingham using dichloromethane/methanol (3:1 v/v) and
ultrasonication. The acid and neutral fractions were separated using an aminopropyl-silica gel column and the
FAs eluted using diethyl ether with 4% acetic acid. The acid fraction was derivatized using boron trifluoride in
methanol and subsequently cleaned up using a silica gel column and the FAs eluted with dichloromethane. FAs
were identified using an Agilent gas chromatograph coupled to an Agilent mass selective detector and
concentrations were quantified using a gas chromatograph – flame ionization detector analysis with the
inclusion of an internal standard ($C_{19}$ alkane) of known concentration. Carbon isotopes were measured with an
Isoprime 100 isotope ratio-mass spectrometer coupled to an Agilent gas chromatograph-flame ionization
detector and a GC5 furnace. Errors are based on the standard deviation of duplicate measures and are all within
0.26‰.
*HBIs*
Two hundred and thirty-four samples were taken every 2 cm over the whole core for highly branched
isoprenoids (HBI) alkenes analysis. HBI were extracted at Laboratoire d'Océanographie et du Climat:
Experimentations et Approches Numériques (LOCEAN), separately from the fatty acids, using a mixture of
9mL $CH_2Cl_2$/MeOH (2:1, v:v) to which internal standards were added and applying several sonication and
centrifugation steps in order to extract properly the selected compounds (Etourneau et al., 2013). After drying
with $N_2$ at 35°C, the total lipid extract was fractionated over a silica column into an apolar and a polar fraction
using 3 mL hexane and 6 mL $CH_2Cl_2$/MeOH (1:1, v:v), respectively. HBIs were obtained from the apolar
fraction by the fractionation over a silica column using hexane as eluent following the procedures reported by
(Belt et al., 2007, Massé et al., 2011). After removing the solvent with $N_2$ at 35°C, elemental sulfur was
removed using the TBA (Tetrabutylammonium) sulfite method (Jensen et al., 1977; Riis and Babel, 1999). The
obtained hydrocarbon fraction was analyzed within an Agilent 7890A gas chomatograph (GC) fitted with 30 m
fused silica Agilent J&C GC column (0.25 mm i.d., 0.25 µm film thickness), coupled to an Agilent 5975C
Series mass selective detector (MSD). Spectra were collected using the Agilent MS-Chemstation software.
Individual HBIs were identified on the basis of comparison between their GC retention times and mass spectra
with those of previously authenticated HBIs (Johns et al., 1999) using the Mass Hunter software. Values are
expressed as concentration relative to the internal standard.





*Diatoms*
One hundred and eighteen samples were taken every 4 cm over the whole core for diatom analyses. Sediment
processing and slide preparation followed the method described in Crosta et al. (2020).
Diatom counting followed the rules described in Crosta and Koç (2007). Around 350 diatom valves were
counted in each sample at a 1000X magnification on a Nikon Eclipse 80i phase contrast microscope. Diatoms
were identified to species or species group level. Absolute abundances of diatoms were calculated following the
equation detailed in Crosta et al. (2008). The relative abundance of each species was determined as the fraction
of diatom species against total diatom abundance in the sample.

**3 Fatty acids within DTGC2011**
Analysis by GC-MS identified seven dominant FAs within the DTGC2011 samples (Fig. S2). These have
carbon chain lengths of $C_{16}$ to $C_{26}$ and only the saturated forms (i.e. no double bonds) were identified. These are
predominantly even chain length FAs, with only minor amounts of the $C_{17}$ compound measured (Gilchrist,

128    2018).

**3.1 Fatty acid concentrations**
Down core analysis of FA concentrations reveals clear groupings in concentration changes. In the upper part of
the core (ca. 3 – 90 cm depth), spanning the last ~78 years, all FA compounds show a similar pattern, with
elevated concentrations, broadly decreasing down-core (Fig. 4). Below this, however, two groups clearly
diverge. These can be broadly divided into short-chained fatty acids ($C_{16}$ to $C_{20}$; SCFAs) and long-chained fatty
acids ($C_{22}$ to $C_{26}$; LCFAs). Within these groups, the concentrations of different compounds show similar trends,
but the two groups (SCFAs vs LCFAs) show different trends to each other (Gilchrist, 2018). This is confirmed
by $R^2$ values calculated for the linear regression of concentrations of each FA against each other throughout the
core (Fig. 2; n = 135, p <0.001). Correlations between the SCFAs have $R^2$ values between 0.97 and 0.99, while
$R^2$ values of LCFAs range between 0.88 and 0.95. Between the two groups, however, $R^2$ values are all lower,
ranging between 0.50 and 0.77.
These distinct groupings suggest that compounds within each group (SCFAs and LCFAs) likely have a common
precursor organism or group of organisms, but the two groups themselves have different producers from each
other. These producers may in turn thrive during different seasons or within different habitats and thus, the
isotopic composition of compounds from these different groups may record different environmental signals.
$R^2$ values were also calculated for samples below 25 cm only, to remove correlations associated with
preservation changes in the top part of the core (discussed below). Although the $R^2$ values are not quite as high,
they broadly confirm these groupings, with the $R^2$ values generally being greater within the two groups (n = 73).
$R^2$ values range from 0.93 for the $C_{18}$ with $C_{20}$, down to 0.07 for the $C_{18}$ and $C_{24}$ (Fig. 3).
The $C_{18}$ and $C_{24}$ FAs are the most abundant compounds within the SCFA and LCFA groups, respectively, and
also the least correlated with each other both in the whole core ($R^2 = 0.5$) and below 25 cm ($R^2 = 0.07$), which
suggests they are the most likely to be produced by different organisms. Furthermore, these two compounds
yielded the highest quality isotope measurements, due to their greater concentrations, clean baseline and





152 minimal coeluting peaks (Fig. S2). Thus, these two compounds ($C_{18}$ and $C_{24}$) will be the focus of analysis and

153 discussion.


### 3.2 Potential sources of the $C_{18}$ fatty acid

156 Potential sources for the $C_{18}$ FA in core U1357 (recovered from the same site as DTGC2011) are discussed in

157 Ashley et al. (*in review*) who suggest the prymnesiophyte *Phaeocystis antarctica* to be the most likely main

158 producer based on a) previous studies (Dalsgaard et al., 2003), b) the high observed abundance of *P. antarctica*

159 within modern Adélie surface waters (Riaux-Gobin et al., 2011) and c) comparison between the measured $\delta^{13}C$

160 values and those reported in the literature for *P. antarctica* (Kopczynska et al., 1995; Wong and Sackett, 1978).

161 Unfortunately, the absence of *P. antarctica* in sediments, as it does not biomineralize any test, precludes the

162 direct comparison of down core trends of this species with FAs. *Phaeocystis antarctica* has been found to live

163 within and underneath sea ice before its break up, as well as in open ocean waters (Riaux-Gobin et al., 2013;

164 Poulton et al., 2007), due to its ability to use a wide range of light intensities for energy production (Moisan and

165 Mitchell, 1999).

166 Dalsgaard et al. (2003) looked at the FAs of eight major microalgal classes and showed that Prymnesiophyceae

167 and Dinophyceae produce the highest proportions of the saturated $C_{18}$ FA, the former to which *P. antarctica*

168 belongs. They also showed that the majority of FAs produced were the unsaturated form which are

169 preferentially broken down in the water column and sediments. As such, although the $C_{18}$ FA represents only a

170 small proportion of the total FA fraction, its higher preservation rate increases its proportion in the sediment.

171 Riaux-Gobin et al. (2011) found *P. antarctica* to dominate the surface waters offshore Adélie Land after spring

172 sea-ice break-up, representing 16% of the phytoplankton assemblage. Although several species of the class

173 Dinophyceae were also recorded, *P. antarctica* was more than 20 times more abundant than the 3 most abundant

174 Dinophyceae taxa combined. Sambrotto et al. (2003) also observed large blooms of *Phaeocystis* sp. in stable,

175 shallow mixed layer water along the edge of fast ice near the Mertz Glacier.

176 Furthermore, Skerratt et al. (1998) identified the FAs produced by *P. antarctica* and two Antarctic diatoms,

177 *Chaetoceros simplex* and *Odontella weissflogii*, from culture samples. Of the FAs produced by *P. antarctica*,

178 52% were saturated FAs ($C_{14}$-$C_{20}$) compared to just 14 and 11% for the two diatoms, respectively, the latter

179 instead producing much more of the mono- and polyunsaturated FAs. The percentage of $C_{18}$ FA produced by *P.*

180 *antarctica* was also 4.1 and 12.5 times greater than the percentage of $C_{18}$ produced by *C. simplex* and *O.*

181 *weissflogii*, respectively. This supports the hypothesis of *P. antarctica* being a dominant and abundant source of

182 the saturated $C_{18}$ FA in the Adélie basin though minor contributions of $C_{18}$ from other phytoplankton species

183 such as the diatoms and dinoflagellates cannot be excluded.

### 3.3 Potential sources of the $C_{24}$ fatty acid

185 Long-chain *n*-alkyl compounds, including FAs, are major components of vascular plant waxes and their

186 presence within sediments has commonly been used as a biomarker of terrestrial plants (Pancost and Boot,

187 2004). Although plants such as bryophytes (e.g. mosses) which are present in the Antarctic do also produce

188 LCFAs (Salminen et al., 2018), it is unlikely that FAs from terrestrial plants make a significant contribution to





the water column, due to their extremely limited extent on the continent, and the significant distance of the site
from other continental sources.
However, there is much evidence in the literature for various aquatic sources of LCFAs, a few of which are
summarized in Table S2. Although not all of these sources are likely to be present within the coastal waters
offshore Adélie Land, it highlights the wide range of organisms which can produce these compounds, and thus
suggests that an autochthonous marine source is entirely possible, especially considering the highly productive
nature of this region.
**3.4 Microbial degradation and diagenetic effects on fatty acid concentration**
Both the $C_{18}$ and $C_{24}$ FAs show an overall decrease in concentrations down-core, with significantly higher
concentrations in the top 80 cm (representing ~70 years) compared to the rest of the core. Below this point, FAs
concentrations variations are attenuated (Fig. 4).
Many studies have shown that significant degradation of FAs occurs both within the water column and surface
sediments as a result of microbial activity, and that there is preferential break down of both short-chained and
unsaturated FA, compared to longer-chained and saturated FA (Haddad et al., 1992; Matsuda, 1978; Colombo et
al., 1997). Haddad et al. (1992) studied the fate of FAs within rapidly accumulating (10.3 cm yr$^{-1}$) coastal
marine sediments (off the coast of North Carolina, USA) and showed that the vast majority (ca. 90%) of
saturated FAs were lost due to degradation within the top 100 cm (representing ~10 years). Similarly, Matsuda
and Koyama (1977) found FA concentrations decrease rapidly within the top 20 cm of sediment (accumulating
at 4 mm yr$^{-1}$) from Lake Suwa, Japan. Assuming similar processes apply to the DTGC2011 sediments, this
suggests the declining concentrations within the upper part of the core are largely the result of diagenetic effects
such as microbial activity occurring within the surface sediments, and thus do not reflect a real change in
production of these compounds in the surface waters.
The complete lack of both unsaturated and short chained (fewer than 16 carbon atoms) FA compounds
identified within DTGC2011 samples, even within the top layers, suggests that selective breakdown of
compounds has already occurred within the water column and on the sea floor (before burial). Wakeham et al.
(1984) assessed the loss of FAs with distance during their transport through the water column at a site in the
equatorial Atlantic Ocean and estimated that only 0.4 to 2% of total FAs produced in the euphotic zone reached
a depth of 389 m, and even less reaching more than 1,000 m depth, the vast majority of material being recycled
in the upper water column. Their results also show a significant preference for degradation of both unsaturated
and short chained compounds over saturated and longer chain length compounds. Although no studies into the
fate of lipids within the water column exist for the Adélie region, the >1,000 m water depth at the core site
would provide significant opportunity for these compounds to be broken down during transportation through the
water column. It is likely, therefore, that the distribution of compounds preserved within the sediments will not
be a direct reflection of production in the surface waters, and explains the preference for saturated FAs with
carbon chain lengths of 16 and more.
Although FA concentrations in the top 80 cm of core DTGC2011 are much higher overall than the sediments
below and show a broad decline over this section, there is a high level of variability. Concentrations do not
decrease uniformly within the top part of the core, as may be expected if concentration change is a first order





response to declining microbial activity. The peak in total FAs instead occurs at a depth of 21-22 cm with a
concentration more than an order of magnitude higher than in the top layer. This variability creates difficulty in
directly determining the effects of diagenesis. However, by 25 cm the concentrations drop to below 1,000 ng g$^{-1}$
and remain so until 32 cm before increasing again. This may suggest that diagenetic effects of FA
concentrations are largely complete by 25 cm (representing ca. 25 years), consistent with results from Haddad et
al. (1992) and Matsuda and Koyama (1977), and that subsequent down-core concentration variations
predominantly represent real changes in export productivity, resulting from environmental factors. However, the
fluctuating nature of concentrations particularly in the youngest sediments means it is difficult to clearly unpick
the effects of diagenesis from actual changes in production of these compounds, and a clear cut-off point for
diagenetic effects cannot be determined.
**3.5 Comparison of fatty acid concentrations with highly branched isoprenoid alkenes**
We compare FA concentrations with other organic compounds (whose source is better constrained) in
DTGC2011 to better understand FA sources. Direct comparison between different organic compound classes
can be made since both are susceptible to similar processes of diagenesis, in contrast to other proxies such as
diatoms. In core DTGC2011, concentrations of di- and tri-unsaturated highly branched isoprenoid (HBI) alkenes
(referred to as HBI diene and HBI triene, respectively hereafter) were available.
In Antarctic marine sediments HBIs have been used as a tool for reconstructing sea ice (Belt et al., 2016, 2017).
Smik et al. (2016) compared the concentrations of HBIs in sediment samples offshore East Antarctica from the
permanently open-ocean zone (POOZ), the marginal ice zone (MIZ) and the summer sea-ice zone (SIZ). They
found the HBI diene reached the highest concentrations in the SIZ and was absent from the POOZ. In contrast,
the HBI triene was most abundant in the MIZ, i.e. at the retreating sea ice edge, with much lower concentrations
in the SIZ and POOZ. This suggests that the two compounds are produced in contrasting environments but
remain sensitive to changes in sea ice.
The HBI diene biomarker (or IPSO$_{25}$ for Ice Proxy Southern Ocean with 25 Carbons) is mainly biosynthesised
by *Berkeleya adeliensis* (Belt et al., 2016), a diatom which resides and blooms within the sea ice matrix, and
thus can be used as a proxy for fast ice attached to the coast. In contrast, the presence of the HBI triene mostly in
the MIZ is suggestive of a predominantly pelagic phytoplankton source (e.g. *Rhizosolenia* spp, Massé et al.,
2011; Smik et al., 2016; Belt et al., 2017), rather than sea-ice dwelling diatoms (Smik et al., 2016). The fact that
HBI triene reached its greatest abundance within the MIZ suggests its precursor organism may thrive in the
stratified, nutrient-rich surface waters of the sea-ice edge.
One key similarity between both the HBI diene and triene, and the FA concentrations is that the highest
concentrations are found in the youngest sediments. These compounds all show broad increases in concentration
from 110 cm depth (ca. 1900 C.E) until the top of the core (Fig. 4 and 5). Concentrations of HBIs are also
susceptible to degradation through the water column through visible light induced photo-degradation (Belt and
Müller, 2013) and diagenetic effects, as well as reacting with sediments resulting in sulphurisation (Sinninghe
Damsté et al., 2007), isomerisation and cyclisation (Belt et al., 2000). Thus, it is likely that the elevated
concentrations, and thus the similarity between FA and HBI concentrations, is due to better preservation at the
top of the core, with diagenetic effects having an increasing and progressive impact down to ca. 25cm depth.





However, despite an overall increase in HBI and FA concentrations above 110 cm depth, there are clear
deviations from this trend. Concentrations of the HBI triene show some broad similarities with FA
concentrations. In particular, both the HBI triene and the $C_{18}$ FA have coeval concentration peaks around 1980-
88, 1967, 1938, 1961-72, 1848 and 1752 C.E. (Fig. 5). These peaks are offset from the HBI diene
concentrations, suggesting that they result from increased production in the surface waters rather than simply
changes in preservation. The HBI triene is more susceptible to degradation than the diene (Cabedo Sanz et al.,
2016), so while this could explain some of the differences between the diene and triene records, where the triene
increases independently of the diene, this is likely to be a genuine reflection of increased production of these
compounds at the surface rather than an artefact of preservation processes.
This close similarity between the $C_{18}$ FA and HBI triene concentrations (Fig. 5) suggests that the $C_{18}$ may also
be produced by an organism associated with the retreating ice edge. *Phaeocystis antarctica* has been proposed
as a potential producer of the $C_{18}$ in core U1357B (Ashley et al., *in review*). In the Ross Sea, *P. antarctica* has
been observed to dominate the phytoplankton bloom during the spring, blooming in deep mixed layers as the sea
ice begins to melt, after which diatoms tend to dominate during the summer (Arrigo et al., 1999; Tortell et al.,
2011; DiTullio et al., 2000). However, a few studies in the Adélie region suggest this is not the case there.
Offshore Adélie Land, *P. antarctica* has been found to only appear late in the spring/early summer, later than
many diatom species. During this time, it occurs preferentially within the platelet ice and under-ice water
(Riaux-Gobin et al., 2013). Furthermore, Sambrotto et al. (2003) observed a surface bloom of *P. antarctica* near
the Mertz Glacier (Fig. 1) during the summer months, in very stable waters along the margin of fast ice and
Riaux-Gobin et al. (2011) found *P. antarctica* to be abundant in the coastal surface waters eight days after ice
break up. This indicates an ecological niche relationship with cold waters and ice melting conditions. This might
explain the close similarity between the $C_{18}$ and HBI triene concentrations, both produced by organisms
occupying a similar habitat at the ice edge.
The $C_{24}$ FA record also shows some similarity with the HBI triene record. This appears to be mostly in the top
part of the core where the highest concentrations are found. The reason for this resemblance is unclear,
especially considering the lack of correlation between the $C_{24}$ and $C_{18}$ FA concentrations. However, it may relate
to the better preservation in younger samples. The weaker coherence between the $C_{24}$ and the HBI triene, and
also HBI diene, suggests that the $C_{24}$ FA is predominantly produced by an organism which is not associated with
sea ice, and thus instead with more open waters.
Seventy-three diatom species were encountered in core DTGC2011 (Campagne, 2015), with *Fragilariopsis*
*curta* and *Chaetoceros* resting spores being the most abundant. However, trends in diatom abundances do not
show any clear correlations with the $C_{18}$ or $C_{24}$ FA concentrations. While this would lend support to the
hypothesis that diatoms are not the main producers of these compounds, the differing effects of diagenesis on
the preservation of diatoms and lipids could also explain some of the differences in observed concentrations,
particularly in the upper part of the core. The known producer of the HBI diene, *Berkeleya adeliensis*, for
example, was not recorded within the core, likely due to their lightly silicified frustules which are more
susceptible to dissolution (Belt et al., 2016). Therefore, despite the lack of a correlation between diatom
abundances and FA concentrations, we cannot entirely rule out the possibility of a minor contribution of FAs by
diatoms.




**4 Carbon isotopes of fatty acids**
Down-core changes in $\delta^{13}C$ for the $C_{18}$ and $C_{24}$ FAs ($\delta^{13}C_{18FA}$ and $\delta^{13}C_{24FA}$, respectively) (Fig. 6 and 7) clearly
show different trends, with very little similarity between them ($R^2 = 0.016$). This further supports the idea that
these compounds are being produced by different organisms, and thus are recording different information.
The mean carbon isotope value of $\delta^{13}C_{18FA}$ of -29.8 ‰ in core U1357 from the same site (Ashley et al., *in*
*review)* is suggestive of a pelagic phytoplankton source (Budge et al., 2008). In core DTGC2011 the mean
values of $\delta^{13}C_{18FA}$ and $\delta^{13}C_{24FA}$ are -26.2 ‰ and -27.6 ‰, respectively. Though more positive, these values are
still within the range of a phytoplankton source. Additionally, the 0.5‰ more positive $\delta^{13}C_{18FA}$ mean value over
the $\delta^{13}C_{24FA}$ may indicate the contribution of sea-ice dwelling algae producers, since carbon fixation occurring
within the semi-closed system of the sea ice will lead to a higher degree of $CO_2$ utilisation than in surrounded
open waters (Henley et al., 2012). Although no studies on FA $\delta^{13}C$ of different organisms are available for the
Southern Ocean, Budge et al. (2008) measured the mean $\delta^{13}C$ value of $C_{16}$ FA from Arctic sea-ice algae (-24.0
‰) to be 6.7 ‰ higher than pelagic phytoplankton (-30.7 ‰) from the same region.
The higher $\delta^{13}C$ of the $C_{18}$ FA could therefore be indicative of *P. antarctica* living partly within the sea ice, e.g.
during early spring before ice break up. The more negative $\delta^{13}C_{24FA}$ suggests it is more likely to be produced by
phytoplankton predominantly within open water.
**4.1 Controls on $\delta^{13}C_{FA}$**
The $\delta^{13}C_{18FA}$ record shows a broadly increasing trend towards more positive values from ca. 1587 until ca. 1920
C.E., with short term fluctuations of up to ~4 ‰ superimposed on this long-term trend (Fig. 7). This is followed
by a period of higher variability with a full range of 5.6 ‰ until the most recent material (ca. 1999 C.E.), with
more negative $\delta^{13}C$ values between 1921 and 1977 C.E. and rapid a shift toward more positive values thereafter.
In contrast, the $\delta^{13}C_{24FA}$ record overall shows a weak, negative trend, with large decadal fluctuations of up to 4.6
‰, with a more pronounced negative trend after ca. 1880 C.E. (Fig. 6 and 7).
Below we consider the various factors which may control the carbon isotope value of algal biomarkers produced
in the surface waters. Down-core changes in FA $\delta^{13}C$ are likely to be a function of either the $\delta^{13}C$ of the
dissolved inorganic carbon (DIC) source, changes in the species producing the biomarkers, diagenesis or
changing photosynthetic fractionation ($\varepsilon_p$). The next section outlines the potential influence of these factors may
have in order to assess the mostly likely dominant driver of FA $\delta^{13}C$.
*4.1.1 Isotopic composition of DIC*
The $\delta^{13}C$ of the DIC source can be affected by upwelling or advection of different water masses, or the $\delta^{13}C$ of
atmospheric $CO_2$. Around the Antarctic, distinct water masses have unique carbon, hydrogen and oxygen
isotope signatures and thus isotopes can be used as water mass tracers (e.g. Mackensen, 2001, Archambeau et
al., 1998). In the Weddell Sea for example, Mackensen (2001) determined the $\delta^{13}C$ value of eight water masses,
which ranged from 0.41 ‰ for Weddell Deep Water, sourced from CDW, to 1.63 ‰ for AASW. A similar
range of ~1.5 ‰ was identified in water masses between the surface and ~5,500 m depth along a transect from
South Africa to the Antarctic coast (Archambeau et al., 1998). Assuming similar values apply to these water
masses offshore Adélie Land, this range in values would be insufficient to explain the ~5 ‰ variation of $\delta^{13}C$





recorded by both $C_{18}$ and $C_{24}$ FA, even in the situation of a complete change in water mass over the core site.
Furthermore, site DTGC2011, located within a 1,000 m deep depression and bounded by the Adélie Bank to the
north, is relatively sheltered from direct upwelling of deep water (Fig. 1). Though inflow of mCDW has been
shown to occur within the Adélie Depression to the east of the bank (Williams and Bindoff, 2003) and possibly
within the Dumont d'Urville Trough, only very small amplitude changes in $\delta^{13}C$ of benthic foraminifera,
tracking upper CDW, have been observed over the Holocene in Palmer Deep, West Antarctica (Shevenell and
Kennett, 2002). Although from a different location, this argues against large changes in the isotopic composition
of the source of mCDW.
Changes in the $\delta^{13}C$ of atmospheric $CO_2$, which is in exchange with the surface waters could also have the
potential to drive changes in the $\delta^{13}C$ of algal biomarkers. Over the last ca. 200 years, the anthropogenic burning
of fossil fuels has released of a large amount of $CO_2$ depleted in $^{13}C$, meaning that the $\delta^{13}C$ of $CO_2$ has
decreased by ca. 1.5 ‰, as recorded in the Law Dome ice core. Prior to this, however, the $\delta^{13}C$ of $CO_2$ in the
atmosphere remained relatively stable, at least for the last thousand years (Francey et al., 1999). Therefore, this
could potentially drive the $\delta^{13}C$ of algal biomarkers towards lighter values within the last 200 years, but this
could not explain the full variation of ~5-6 ‰ in FA $\delta^{13}C$ measured throughout the core. No clear trend towards
lighter values is evident in the last 200 years of the FA $\delta^{13}C$ records, which suggests that this change is
insignificant compared to local and regional inter-annual variations as a result of other environmental drivers
(discussed below).
*4.1.2 Changing species*
A shift in the organisms producing the FA could also affect $\delta^{13}C$ where species have different fractionation
factors. For example, changing diatom species have been shown to have an effect on bulk organic matter $\delta^{13}C$ in
core MD03-2601, offshore Adélie Land, over the last 5 ka (Crosta et al., 2005). However, the bulk organic
matter might have contained other phytoplankton groups than diatoms with drastically different $\delta^{13}C$ values and
fractionation factors. Here we measured $\delta^{13}C$ of individual biomarkers, produced by a more restricted group of
phytoplankton groups (possibly restricted to a few dominant species) compared to bulk $\delta^{13}C$. As discussed
above, the $C_{18}$ appears to be produced predominantly by *P. antarctica,* whereas diatoms do not tend to produce
this compound (Dalsgaard et al., 2003).
*4.1.3 Effect of diagenesis on lipid $\delta^{13}C$*
Sun et al. (2004) studied the carbon isotope composition of FAs during 100 days of incubation in both oxic and
anoxic seawater. They observed a shift towards more positive values in FA $\delta^{13}C$, ranging between 2.6 ‰ for the
$C_{14:0}$ and as much as 6.9‰ in the $C_{18:1}$, under anoxic conditions. This suggests that diagenesis could affect FA
$\delta^{13}C$ in core DTGC2011. However, these observed changes are rapid (days to months), occurring on timescales
which are unresolvable in the FA $\delta^{13}C$ record (annual to decadal), and thus may have no effect on the trends
observed in our record. Based on concentration data discussed above, it seems that diagenetic overprint is
largely complete by ~25 cm (Fig. 4). In the top 25 cm of the core, the $\delta^{13}C_{24FA}$ values increase by ~2.5 ‰,
downward ($R^2 = 0.63$, n = 11) while the $\delta^{13}C_{18FA}$ values display a large variation with no overall trend ($R^2 =$
0.12, n = 20). If diagenesis was driving the changes in $\delta^{13}C$, it is likely that this trend would be observed in all
FA compounds.



Taken together, it appears that neither changes in the $\delta^{13}C$ of the DIC, changing phytoplankton groups nor
diagenesis can fully explain the variation of FA $\delta^{13}C$ recorded within DTGC2011. Therefore, we hypothesise
that changes in $\varepsilon_p$ are the main driver of FA $\delta^{13}C$.
**4.2 Controls on photosynthetic fractionation ($\varepsilon_p$)**
There is a positive relationship between $\varepsilon_p$ in marine algae and dissolved surface water $CO_{2(aq)}$ concentration
(Rau et al., 1989). As a result, higher $\delta^{13}C$ values are hypothesised to reflect lower surface water $CO_{2(aq)}$ and vice
versa. Changes in surface water $CO_{2(aq)}$ concentration in turn may be driven by various factors, including
changing atmospheric $CO_2$ (Fischer et al., 1997), wind-driven upwelling of deep, carbon-rich water masses
(Sigman and Boyle, 2000; Takahashi et al., 2009), sea-ice cover (Henley et al., 2012) and/or primary
productivity (Villinski et al., 2008). Thus, determining the main driver(s) of surface water $CO_2$ changes offshore
Adélie Land should enable interpretation of the DTGC2011 FA $\delta^{13}C$ records.
*4.2.1 Sea ice*
Brine channels within sea ice have very low $CO_2$ concentrations and a limited inflow of seawater. Carbon
isotopic fractionation of algae living within these channels has been shown to be greatly reduced compared to
organisms living in the surrounding open waters (Gibson et al., 1999), leading to elevated $\delta^{13}C$ values. It is thus
possible that, under conditions of high sea-ice cover, enhanced FA contribution from sea-ice algae leads to
elevated sedimentary $\delta^{13}C$ values. HBI diene concentrations within DTGC2011 show a much greater presence
of fast ice at the core site ca. 1960 C.E (Fig. 5). However, during this time there is no clear elevation in $\delta^{13}C$
concentrations in either $\delta^{13}C_{18FA}$ or $\delta^{13}C_{24FA}$, both instead showing generally lower $\delta^{13}C$ values. In fact, $\delta^{13}C_{18FA}$
shows the lowest values of the whole record between 1925 and 1974 C.E., during which sea ice, as recorded by
the HBI diene, is at its highest level. This suggests that inputs in sea-ice algae at this time are not driving
changes in FA $\delta^{13}C$.
The DTGC2011 core site sits proximal to the Dumont D'Urville polynya, which has a summer area of 13.02 x
$10^3$ km$^2$ and a winter area of 0.96 x $10^3$ km$^2$ (Arrigo and van Dijken, 2003). Changes in the size of the polynya
both on seasonal and inter-annual time scales will affect air-sea $CO_2$ exchange and thus also surface water $CO_2$
concentration. A reduced polynya may lead to greater supersaturation of $CO_2$ in the surface waters due to
reduced outgassing, allowing $CO_2$ to build up below the ice, leading to lower $\delta^{13}C$ values of algal biomarkers
produced in that habitat (Massé et al., 2011). Thus changes in the extent of sea ice may also effect FA $\delta^{13}C$.
*4.2.2 Observed trends in surface water $CO_{2(aq)}$*
If the trend in surface water $CO_{2(aq)}$ paralleled atmospheric $CO_2$, with an increase of over 100 ppm over the last
200 years (MacFarling Meure et al., 2006), we might expect phytoplankton to exert a greater fractionation
during photosynthesis in response to elevated surface water $CO_{2(aq)}$ concentration, resulting in more negative
$\delta^{13}C$ values. Taking into account the decline in atmospheric $\delta^{13}CO_2$ over the same period would further enhance
the reduction in phytoplankton $\delta^{13}C$. Fischer et al. (1997) looked at the $\delta^{13}C$ of both sinking matter and surface
sediments in the South Atlantic and suggested that, since the preindustrial, surface water $CO_{2(aq)}$ has increased
much more in the Southern Ocean than in the tropics. They estimated that a 70 ppm increase in $CO_{2(aq)}$ in
surface waters of 1°C would decrease phytoplankton $\delta^{13}C_{org}$ by ca. 2.7‰, and up to 3.3‰ $\delta^{13}CO_2$ change are
included, between preindustrial and 1977-1990. However, sea ice cover and summer primary productivity are





likely to be much higher off Adélie Land than in the South Atlantic, both of which will affect air-sea gas
exchange.
Shadwick et al. (2014) suggest that surface water $CO_2$ should track the atmosphere in the Mertz Polynya region,
despite the seasonal ice cover limiting the time for establishing equilibrium with the atmosphere. They
calculated wintertime $CO_2$ in the shelf waters of the Mertz Polynya region, offshore Adélie Land (Fig. 1),
measuring ca. 360 ppm in 1996, ca. 396 ppm in 1999, and ca. 385 ppm in 2007, while atmospheric $CO_2$ at the
South Pole was 360, 366 and 380 ppm, respectively (Keeling et al., 2005). Based on the 1996 and 2007 data
only, an increase in $CO_2$ of ca. 25 ppm is observed over these 11 years, coincident with the 20 ppm atmospheric
$CO_2$ increase over this time period. However, high interannual variability (± ca. 30 ppm) is evident (e.g. 396
ppm in 1999) suggesting that other factors, particularly upwelling, may override this trend. The latter was also
suggested by Roden et al. (2013) based on winter surface water measurements in Prydz Bay, indicating that
decadal-scale carbon cycle variability is nearly twice as large as the anthropogenic $CO_2$ trend alone.
During the austral winter, upwelling of deep water masses causes $CO_2$ to build up in the surface waters, and sea
ice cover limits gas exchange with the atmosphere (Arrigo et al., 2008; Shadwick et al., 2014). Although only
limited data, the measurements by Shadwick et al. (2014) suggest slight supersaturation, of up to 30 ppm, occurs
in the winter due to mixing with carbon-rich subsurface water, but with high interannual variability. This is
compared to undersaturation of 15 to 40 ppm during the summer as a result of biological drawdown of $CO_2$.
Roden et al. (2013) also observed varying levels of winter supersaturation in Prydz Bay, East Antarctica, with
late winter $CO_2$ values of 433 ppm in 2011 (45 μatm higher than atmospheric $CO_2$), and suggested that
intrusions of C-rich mCDW onto the shelf may play a part in this. Similarly, winter surface water $CO_2$ of 425
ppm has been measured by Sweeney (2003) in the Ross Sea, before being drawn down to below 150 ppm in the
summer as phytoplankton blooms develop.
Enhanced upwelling of deep carbon-rich waters in the Southern Ocean are thought to have played a key role in
the deglacial rise of atmospheric $CO_2$, increasing $CO_2$ concentrations by ~80 ppm (Anderson et al., 2009; Burke
and Robinson, 2012). Changes in upwelling offshore Adélie Land could therefore drive some interannual
variability in surface water $CO_2$ and hence FA $\delta^{13}C$ in DTGC2011. However, upwelling tends to be stronger
during the winter months, when sea-ice formation and subsequent brine rejection drive mixing with deeper C-
rich waters. At this time, heavy sea-ice cover limits air-sea gas exchange and enhances $CO_2$ supersaturation in
regional surface waters (Shadwick et al., 2014). In contrast, the phytoplankton producing FA thrive during the
spring and summer months during which $CO_2$ is rapidly drawn down and the surface waters become
undersaturated. However, upwelling cannot be discarded as a possible contributor to surface water $CO_2$ change.
However, the core site is in a relatively sheltered area and is probably not affected by significant upwelling.
Based on these studies, changes in atmospheric $CO_2$ concentration and $\delta^{13}C$ of the source appear to be unlikely
to be a dominant driver of the FA $\delta^{13}C$ record, with interannual variations driven by other factors overriding any
longer-term trend. There is also no clear anthropogenic decline in the FA $\delta^{13}C$ record over the last 200 years,
which supports this hypothesis.
*4.2.3 Productivity*



Given that changes in atmospheric $CO_2$, source signal, sea ice algae or diagenesis seem unable to explain the
full range of variability seen in the FA $\delta^{13}C$ record, the most plausible driver appears to be changes in surface
water primary productivity. Coastal polynya environments in the Antarctic are areas of very high primary
productivity (Arrigo and van Dijken, 2003). The DTGC2011 core site sits near to the Dumont D'Urville
polynya, and is just downstream of the larger and more productive MGP (Arrigo and van Dijken, 2003). In large
polynyas such as the Ross Sea, primary productivity leads to intense drawdown of $CO_2$ in the surface waters,
resulting in reduced fractionation by the phytoplankton during photosynthesis (Villinski et al., 2008). In the
Ross Sea, surface water $CO_2$ has been observed to drop to below 100 ppm during times of large phytoplankton
blooms (Tortell et al., 2011) demonstrating that primary productivity can play a key role in controlling surface
water $CO_2$ concentrations in a productive polynya environment. Arrigo et al. (2015) found the MGP to be the 8th
most productive polynya in the Antarctic (out of 46) based on total net primary productivity during their
sampling period, and Shadwick et al. (2014) observed $CO_2$ drawdown in the MGP during the summer months.
Therefore, we suggest that FA $\delta^{13}C$ signals recorded in DTGC2011 is predominantly a signal of surface water
$CO_2$ driven by primary productivity. Indeed, the potential for the $\delta^{13}C$ of sedimentary lipids to track surface
water primary productivity has been recognised in the highly productive Ross Sea polynya. High variability in
surface water $CO_2$ values have been measured across the polynya during the summer months (December –
January), ranging from less than 150 ppm in the western Ross Sea near the coast, to >400 ppm on the northern
edge of the polynya. This pattern was closely correlated with diatom abundances, indicating intense drawdown
of $CO_2$ in the western region where diatom abundances were highest (Tortell et al., 2011). This spatial variation
in productivity is recorded in particulate organic carbon (POC) $\delta^{13}C$, and is also tracked in the surface sediments
by total organic carbon (TOC) $\delta^{13}C$ and algal sterol $\delta^{13}C$, all of which show significantly higher values in the
western Ross Sea. This spatial pattern in sterol $\delta^{13}C$ was concluded to be directly related to $CO_2$ drawdown at
the surface, resulting in average sterol $\delta^{13}C$ values varying from -27.9‰ in the west, where productivity is
greatest, down to -33.5‰ further offshore (Villinski et al., 2008).
A similar relationship is evident in Prydz Bay, where POC $\delta^{13}C$ was found to be positively correlated with POC
concentration and negatively correlated with nutrient concentration, indicating greater drawdown of $CO_2$ and
nutrients under high productivity levels (Zhang et al., 2014).
This suggests it is possible to apply FA $\delta^{13}C$ as a palaeoproductivity indicator in the highly productive Adélie
polynya environment. However, it is important to constrain the most likely season and habitat being represented,
since phytoplankton assemblages vary both spatially (e.g. ice edge or open water) and temporally (e.g. spring or
summer). The incredibly high sedimentation rate (1-2 cm yr$^{-1}$) within the Adélie Basin is thought to result, on
top of regional high productivity, from syndepositional focusing processes bringing biogenic debris from the
shallower Adélie and Mertz banks to the ca. 1,000 m deep basin (Escutia et al., 2011). Thus, it is likely that core
DTGC2011 contains material from a wide area, including both the Mertz and Dumont d'Urville polynyas, and
areas both near the coast and further offshore, meaning it is quite possible that the $C_{18}$ and $C_{24}$ FAs are
integrating palaeoproductivity changes weighted towards different regional environments, which would explain
their different trends. Furthermore, surface water $CO_2$ can vary spatially, such as in the Ross Sea polynya where
Tortell et al. (2011) measured surface water $CO_2$ values ranging between 100 and 400 ppm. Thus, it is likely





that these two areas offshore Adélie Land where the $C_{18}$ and $C_{24}$ FAs are being produced will also have differing
surface water $CO_2$ concentrations and trends.

**4.3 Comparison of fatty acid $\delta^{13}C$ with other proxy data**

Comparison of down-core variations in FA $\delta^{13}C$ with other proxy data can also be used to decipher the main
signal recorded. Comparison between $\delta^{13}C_{24FA}$ and the major diatom species abundances within the core shows
a reasonably close coherence with *Fragilariopsis kerguelensis,* particularly since ~1800 C.E. (Fig. 6).
*Fragilariopsis kerguelensis* is an open water diatom species and one of the most dominant phytoplankton
species offshore Adélie Land (Chiba et al., 2000), reaching its peak abundance in the summer (Crosta et al.,
2007). This suggests that the $C_{24}$ FA is being produced during the summer months and, as such, is reflecting
productivity in more open waters. The $\delta^{13}C_{24FA}$ record does not show any similarity to the sea-ice records, as
inferred by HBI diene concentrations and abundances of *Fragilariopsis curta* (Fig. 6 and 7), here again
suggesting that these compounds are being produced in open water during the summer months after sea ice has
retreated.
As discussed above, *P. antarctica* is a likely producer for the $C_{18}$ FA, a prymnesiophyte algae which has been
observed in the Adélie region in summer months residing predominantly along the margin of fast ice, but also
further offshore (Riaux-Gobin et al., 2013, 2011; Vaillancourt et al., 2003). The aversion of *F. kerguelensis* to
sea ice (and thus also the $C_{24}$ FA producer) in contrast to *P. antarctica*, may explain the clear lack of coherence
in the down-core trends in $\delta^{13}C_{18FA}$ and $\delta^{13}C_{24FA}$ (Fig. 7). Thus, we hypothesise that $\delta^{13}C_{18FA}$ is recording surface
water $CO_2$ driven by productivity in the MIZ, whilst $\delta^{13}C_{24FA}$ is recording surface water $CO_2$ in more open
water, further from the sea-ice edge.
HBI diene concentrations indicate elevated fast ice cover between ~1919 and 1970 C.E., with a particular peak
between 1942 and 1970 C.E., after which concentrations rapidly decline and remain low until the top of the core
(Fig. 7). Abundances of *F. curta*, used as a sea-ice proxy, similarly show peaks at this time indicate increased
sea-ice concentration (Campagne, 2015) (Fig. 7). $\delta^{13}C_{18FA}$ indicates a period of low productivity between ~1922
and 1977 C.E., broadly overlapping with this period of elevated fast ice concentration (Fig. 7), with a mean
value of -27.12‰. This is compared to the mean value of -26.23‰ in the subsequent period (~1978 to 1998
C.E.) during which HBI diene concentration remain low (Fig. 7). This suggests that productivity in the coastal
region was reduced, while sea-ice concentrations were high. This might be expected during a period of
enhanced ice cover – perhaps representing a reduction in the amount of open water, or a shorter open water
season – since the majority of productivity generally takes place within open water (Wilson et al., 1986).
Furthermore, $\delta^{13}C_{18FA}$ shows a broad similarity with *Chaetoceros* resting spores (CRS) on a centennial scale,
with lower productivity at the start of the record, ca. 1587 to 1662 C.E., followed by an increase in both proxies
in the middle part of the record, where $\delta^{13}C_{18FA}$ becomes relatively stable and CRS reaches its highest
abundances of the record. This is then followed in the latter part of the record, after ca. 1900 C.E., by both
proxies displaying lower values overall. CRS are associated with high nutrient levels and surface water
stratification along the edge of receding sea ice, often following high productivity events (Crosta et al., 2008).
The broad similarity to CRS, with lower values recorded during periods of high sea-ice concentrations, suggests



that $\delta^{13}C_{18FA}$ is similarly responding to productivity in stratified water at the ice edge. This supports the
hypothesis that $\delta^{13}C_{18FA}$ is recording primary productivity in the MIZ.
**5 Conclusions**
FAs identified within core DTGC2011, recovered from offshore Adélie Land, were analysed for their
concentrations and carbon isotope compositions to assess their utility as a palaeoproductivity proxy in an
Antarctic polynya environment. The $C_{18}$ and $C_{24}$ compounds yielded the best isotope measurements and show
very different $\delta^{13}C$ trends, suggesting they are being produced by different species in different habitats and/or
seasons.
Comparison with other proxy data and information from previous studies suggests that the $C_{18}$ compound may
be predominantly produced by *P. antarctica*, with $\delta^{13}C_{18FA}$ reflecting productivity changes in the marginal ice
zone, where it is sensitive to changes in ice cover. In contrast, $\delta^{13}C_{24FA}$, which compares well with abundances
of the open water diatom *F.s kerguelensis,* may be reflecting summer productivity further offshore, in open
waters where it is less sensitive to fast ice changes. We argue that FA $\delta^{13}C$ can be used as a productivity proxy,
but should be used in parallel with other proxies such as diatoms abundances or HBIs. The use of $\delta^{13}C$ analysis
of multiple FA compounds, as opposed to individual compounds or bulk isotope analysis, allows a more
detailed insight into the palaeoproductivity dynamics of the region, with the potential to separate productivity
trends within different habitats.
However, there are clearly uncertainties in interpreting the FA $\delta^{13}C$, and although we have made parsimonious
interpretations, many assumptions have been made here. The producers of the $C_{18}$ and especially the $C_{24}$ FAs is
a key source of uncertainty and will require further work to further elucidate. The possibility of inputs of FAs
from multiple sources, in particular from organisms further up the food chain, has consequences for their
interpretation since this could mean the $\delta^{13}C$ FA is not fully reflecting just surface water conditions. Other key
uncertainties are the magnitude of upwelling of $CO_2$ at the site in comparison to drawdown by phytoplankton,
and the potential role of changes in air-sea $CO_2$ exchange.

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

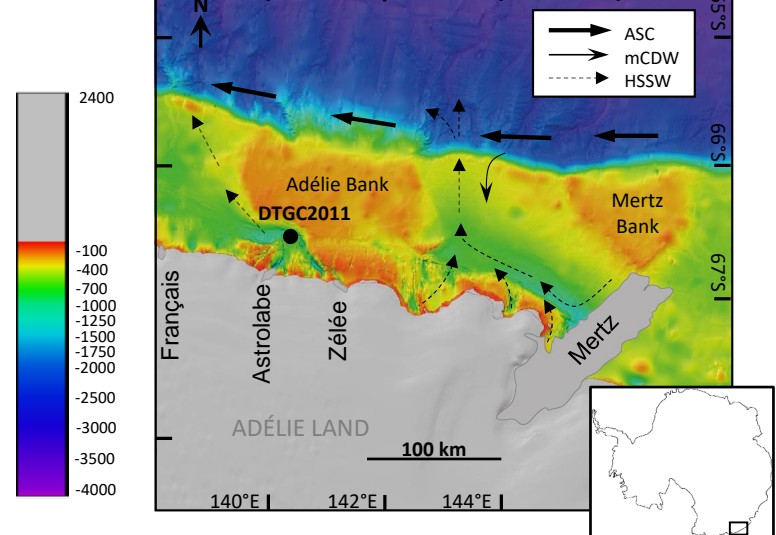

**Figure 1: Location of Site DTGC2011 on bathymetric map of the Adélie Land region (modified from Beaman et al., 2011), indicating positions of the main glaciers (prior to Mertz Glacier Tongue collapse in 2010) and pathways of the main water masses affecting the region: Antarctic Slope Current (ASC), Modified Circumpolar Deep Water (mCDW) and High Shelf Salinity Water (HSSW) (Williams and Bindoff, 2003).**









|  | $C_{16}$ | $C_{17}$ | $C_{18}$ | $C_{20}$ | $C_{22}$ | $C_{24}$ | $C_{26}$ |
|---|---|---|---|---|---|---|---|
| $C_{16}$ |  | 0.97 | 0.98 | 0.97 | 0.72 | 0.53 | 0.58 |
| $C_{17}$ | 0.97 |  | 0.96 | 0.96 | 0.70 | 0.52 | 0.56 |
| $C_{18}$ | 0.98 | 0.96 |  | 0.99 | 0.69 | 0.50 | 0.55 |
| $C_{20}$ | 0.97 | 0.96 | 0.99 |  | 0.77 | 0.59 | 0.64 |
| $C_{22}$ | 0.72 | 0.70 | 0.69 | 0.77 |  | 0.88 | 0.95 |
| $C_{24}$ | 0.53 | 0.52 | 0.50 | 0.59 | 0.88 |  | 0.90 |
| $C_{26}$ | 0.58 | 0.56 | 0.55 | 0.64 | 0.95 | 0.90 |  |

Key:
| 0.9-0.99 |
| 0.8-0.89 |
| 0.7-0.79 |
| 0.6-0.69 |
| 0.5-0.59 |


**Figure 2: $R^2$ values for fatty acid concentrations throughout core DTGC2011. Values are colour coded according to the key on the left. Black border denotes correlations within each group**.





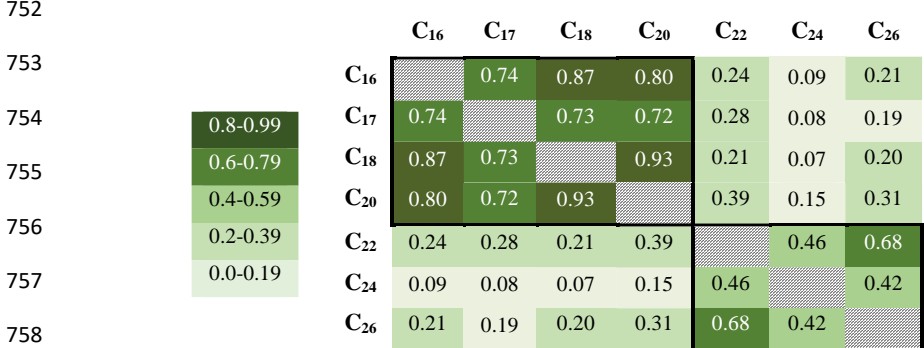

**Figure 3: $R^2$ values for fatty acid concentrations in core DTGC2011 below 25 cm only. Values are colour coded according to the key on the left. Black border denotes correlations within each group.**


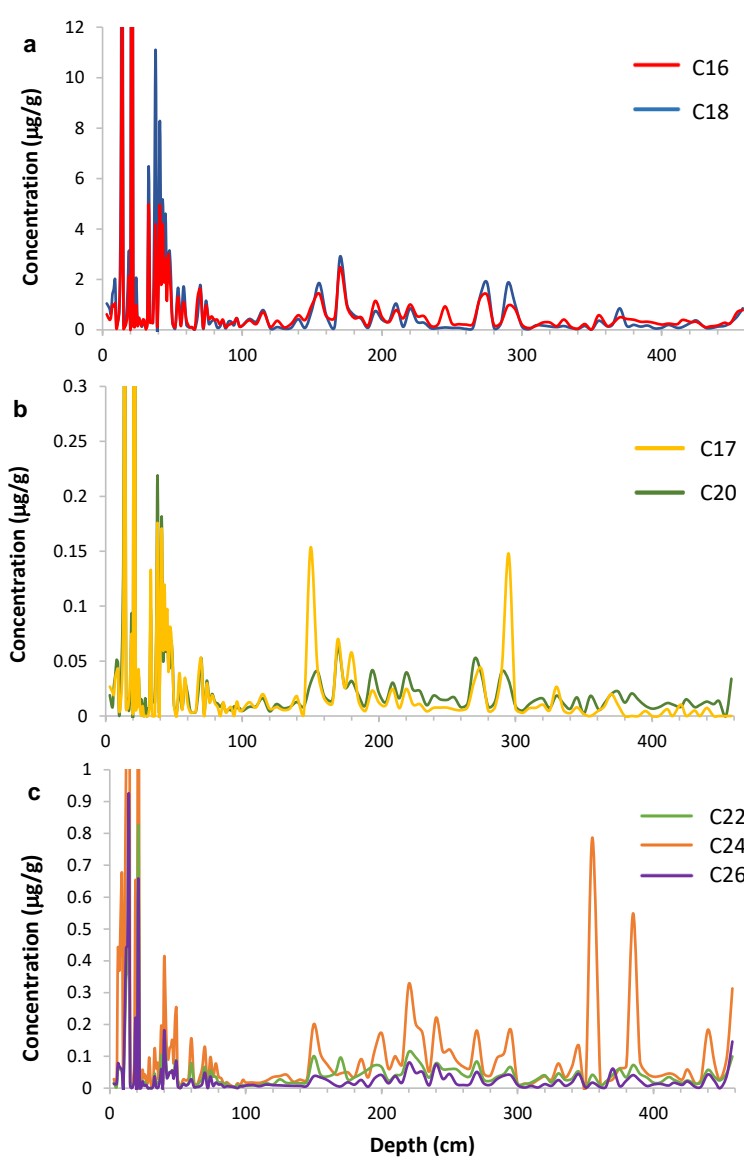

**Figure 4: Fatty acid concentrations (µg/g of dry sediment) with depth from core DTGC2011
a) C₁₆ and C₁₈ fatty acids b) C₁₇ and C₂₀ fatty acids c) C₂₂, C₂₄ and C₂₆ fatty acids.**






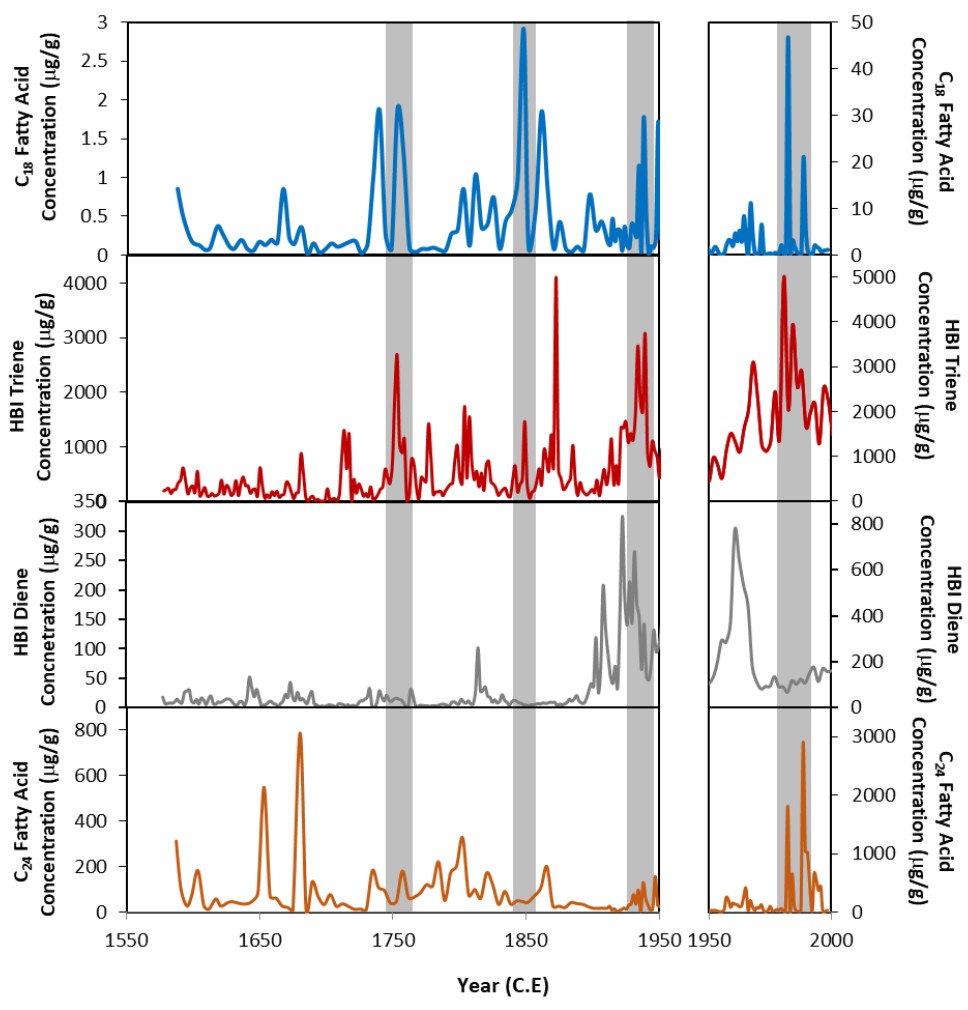

**Figure 5: Concentrations of the C$_{18}$ fatty acid (blue), the HBI triene (red), HBI diene (grey) (Campagne, 2015), C$_{24}$ fatty acid (orange) from core DTGC2011. The left-hand panels show 1550 to 1950 C.E. and the right hand panels show 1950 to 2000 C.E., plotted on different y-axes due to the elevated concentrations in the top part of the core. Grey vertical bands highlight coincident peaks in C$_{18}$ fatty acid and HBI triene records.**








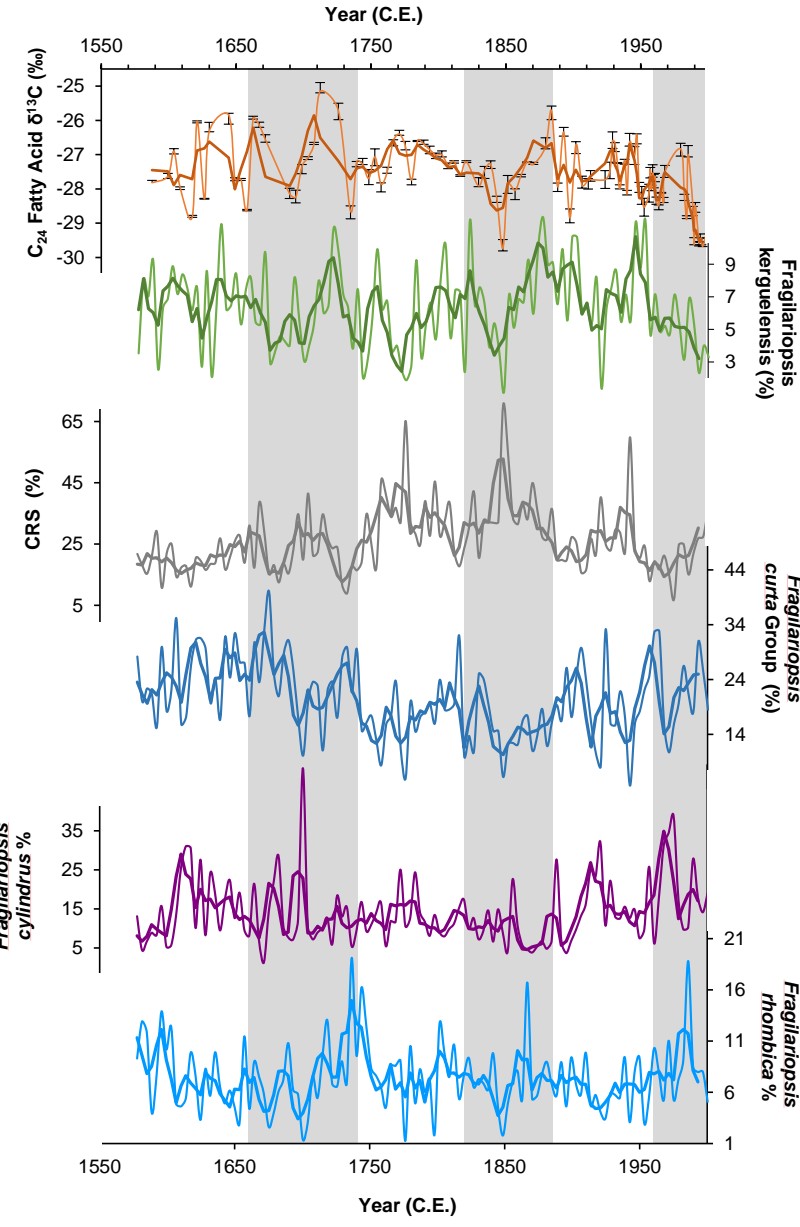

**Figure 6: δ¹³C values of the C₂₄ fatty acid (orange) and relative abundances (%) of the open water diatom**
***Fragilariopsis kerguelensis*** **(green). Also shown are relative abundances of the four most abundant diatom**
**groups in DTGC2011.** ***Chaetoceros*** **resting spores (CRS; grey line),** ***Fragilariopsis curta*** **group (dark blue line),**
***Fragilariopsis cylindrus*** **(purple line) and** ***Fragilariopsis rhombica*** **(light blue line). Thick line represents 3-point**
**moving average for each. Grey vertical bands highlight periods where C₂₄ fatty acid δ¹³C is in phase with** ***F.***
***kerguelensis***.


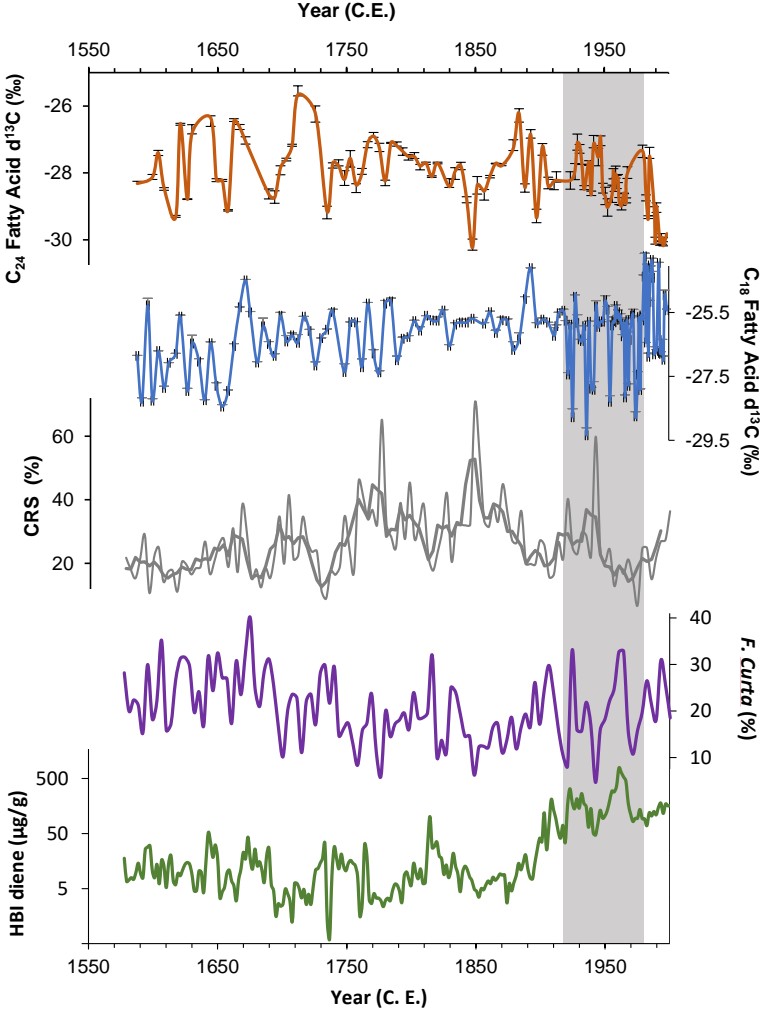

**Figure 7: δ¹³C of the C$_{24}$ (orange) and C$_{18}$ (blue) fatty acid, HBI diene concentrations (green; plotted on a log scale) and relative abundances of *Fragilariopsis curta* plus *Fragilariopsis cylindrus* (purple). Latter two records reflect sea ice concentrations. Grey vertical band highlights period where low C$_{18}$ δ¹³C overlaps with elevated HBI diene concentrations.**

