# Peer review of "Exploring the use of compound-specific carbon isotopes as a"

_Biogeosciences, 2020_

## Referee Comment (RC1) · Anonymous Referee #1 · 26 May 2020

Ashley et al. present an assessment on the usefulness of d13C of fatty acids to assess paleoproductivity in an Antarctic coastal setting The manuscript is well-written, the data appropriate and extensive, and the research question interesting and relevant. The rationale for this work is fully explained, the introduction is clear and the methodology is sound. The main results and discussion section is generally clear, but not enough attention and focus is given to linking the data to productivity. At present, it almost looks like productivity was chosen because the trends could not be explained by anything else. I am sure this is not the case, but it needs to be made clearer for the reader as well.

There are a few criticisms I have which ought to be addressed before this manuscript is ready for publication.

[Figure]

1. The manuscript is focusing on one specific site, and while the observed links to productivity are observed here, the site is very particular and in no way is this ready to be extrapolated at all to any other sites in Antarctica or any other settings. Hence, the title is a little presumptuous, while at the same time the phrasing as a question makes it vague. The phrasing of "fatty acid carbon isotopes" won't be valued by some in the isotope community as it can sound a little bit colloquial. I would suggest changing to "d13C of fatty acids trace paleoproductivity off the coast of Adelie Land, Antarctica" or something along these lines. 2. The manuscript gives a lot of space for trying to pin down a single, or majority, producer, for fatty acids such as C18. I think this is impossible as so many organisms produce C18 FA, and thus this discussion can be shortened and focused. 3. The changes observed in d13C are very small and some comments on how significant changes of 1‰ really are would be useful. 4. I can see a number of analytical issues that should be addressed. First of all, there is no explanation on how the correction for the methyl-group 13C values was carried out. This needs to be explained, or, if the C used for methylation has not been analysed for 13C and is not available anymore, and it is thus impossible to make this correction, it needs to be clearly acknowledged that values are not absolute. The second issue is that the standard used (C19) is not the best for FAME as it is an n-alkane, and was only added post-extraction, hence analysis is semi-quantitative at best which needs to made clearer. 5. Throughout the manuscript, often words such as "extremely", "very high", etc. are used – I would recommend a thorough edit removing these descriptions and replacing them with actual values that allow the reader to put them into context.

Line 68: Give a number instead of "extremely high" – how high? Line 70: "highly productive" as above Line 94: See comment 4 on internal standard – when was it added? Does it really allow quantification at this point? Line 97: Are these values corrected for Me? Are these errors subsequently appropriately propagated? What is the significance of a change of just above 3 x SD (0.26 vs 1 ‰)? Line 102: Which internal standards? Line 194: Saying that a marine source is "entirely possible" sounds strange – do you want to say likely? Lines 213-214: There are more novel studies on FA, Wakeham and

also Hilary Close Line 291: What do you mean by weaker coherence? Lines 547-549: We know that there are many algae that make these FA so this is not likely to be resolved. At the same time, the non-distinctive nature of these molecules will make it difficult to apply this proxy to other settings where there are likely other producers. The whole paragraph is not particularly relevant and I would shorten and/or delete or move up so the work does not finish on a weak statement.

---

## Referee Comment (RC2) · Anonymous Referee #2 · 4 Jun 2020

The high latitude region of the Southern Hemisphere which include Antarctic ice sheet and Southern Ocean is thought to play an important role in climate system, especially in long-climate change. Hence, it is important to investigate paleoclimate change the region to better understand Earth's climate. However, due to limited application of environmental proxies in the region, significant portions of Earth history, environmental records in the high latitude region are less developed than that of low and mid latitudes. Lower and higher molecular fatty acids that are produced by varieties of organisms in the ocean environment are ubiquitous in ocean sediments. Thus, fatty acids may have a potential as paleoenvironmental proxy. This study explores paleoclimatic utility of fatty acids in Southern Ocean sediments and suggests that stable carbon isotope ratio of the low (C18) and mid (C24) chain fatty acids could be used as productivity proxy in the

sea ice area. Although further studies are needed to confirm robustness of the proxy, this study contributes development of biogeochemical proxy which has a potential to apply to high latitude ocean sediment. Hence, this study fits scope of Biogeosciences and suitable for publication in the journal. I have some comments on the article as below.

1. I would suggest to include some explanations that application of biomarker proxy is limited in polar regions into the introduction section (e.g. a powerful proxy such alkenone is not applicable in this region. HBI compounds, that are useful proxy of sea ice, are labile and cannot be applied to geological deep past. On the other hand, fatty acids are ubiquitous and abundantly detected even in old sediment and has a potential but its utility has not been investigated well). Such explanations highlight importance of this work. 2. Although a number of fatty acids including C16 to C26 were abundantly detected in the studied samples (Figure S2), the authors show and discuss d13C results of C18 and C24 fatty acids only. I wonder why the authors focus the two compounds only. I suppose that aim of this paper is to investigate paleoclimatic utility of fatty acids in marine sediments. Hence, it is worth to also include results of the other compounds into the manuscript. I think many people are interested in results of other compounds and know how d13C profiles of other compounds look like. Including this significantly contributes to develop application of fatty acids in marine sediments to paleoclimate study. 3. As for pCO2 effect on plankton d13C, important literature is missing in the manuscript (Pop et al., 1999, vol 13, 827-843, GBC). They measured d13C of POC along the north-south transect of the Southern Ocean and show significant negative correlation between dissolved CO2 and d13C of POC, suggesting strong control of pCO2 on d13C of phytoplankton. There needs to take into consideration the result for discussion. 4. 4.2.3. Productivity section: The authors argue that changes in productivity is the most plausible driver for variability of fatty acid d13C recorded in the sediment core based on the results of previous studies conducted in the Southern Ocean (Villinski et al., 2008; Arrigo et al., 2015; Zhang et al., 2014). I basically agree that significant increase in productivity results in remarkable higher values of

[Figure]

phytoplankton d13C in the polynya environment. However, those papers (Villinski et al., 2008; Arrigo et al., 2015; Zhang et al., 2014) all argue that observed increases in productivity in the regions are caused by meltwater input which promote surface stratification in summer time with reducing vertical mixing and supplying Fe, providing ideal condition for algal growth. Shadwick et al., GRL (2013) and Jack Pan et al., PlosOne (2019) also clearly show a significant correlation between meltwater fraction, chlorophyll concentration and surface water pCO2 drawdown. Especially, Shadwick et al., GRL (2013) investigates glacial meltwater impact on biological carbon drawdown in the studied region. Indeed, those paper shows lowering surface pCO2 happened in the regions where meltwater plume intruded. Regardless of sea ice fluctuations, plankton production takes place in summer when ice sheet melts. This suggests variability of meltwater input rather significantly affects productivity. Therefore, I would suggest to consider possible link between meltwater and productivity in the manuscript. Indeed, the observed resent increase in d13C of C16 fatty acid in sediment core is consistent with the fact of significant melting of Antarctic ice sheet for the past decades. 5. F. cylindrus% and F. rhombica% records are shown in Figures 6 and 7, but the authors do not mention anything about those records in the manuscript. I wonder why those data are shown in the figures.

---

## Author Comment (AC1) · 2 Jul 2020

**Response to reviewer 1 (our replies are in bold)**

Ashley et al. present an assessment on the usefulness of d13C of fatty acids to assess paleoproductivity in an Antarctic coastal setting The manuscript is well-written, the data appropriate and extensive, and the research question interesting and relevant. The rationale for this work is fully explained, the introduction is clear and the methodology is sound. The main results and discussion section is generally clear, but not enough attention and focus is given to linking the data to productivity. At present, it almost looks like productivity was chosen because the trends could not be explained by anything else. I am sure this is not the case, but it needs to be made clearer for the reader as well.

**There are numerous processes affecting carbon isotopes in organic matter, so interpreting the algal $\delta^{13}C$ signal in marine sediments is never going to be an easy task. We therefore feel it is best practise to explore all these possible factors, in the context of the Antarctic polynya environment, before we can conclude, parsimoniously, that productivity is the most likely driver. Various previous studies have shown that productivity is an important driver of $\delta^{13}C$ of organic matter in similar high productivity environments (e.g. the Ross Sea) and this was a starting point for the work. However, we wanted to consider a range of potential drivers. In our final submission, we will therefore add some text at the start of the discussion to explain this.**

There are a few criticisms I have which ought to be addressed before this manuscript is ready for publication.

1. The manuscript is focusing on one specific site, and while the observed links to productivity are observed here, the site is very particular and in no way is this ready to be extrapolated at all to any other sites in Antarctica or any other settings. Hence, the title is a little presumptuous, while at the same time the phrasing as a question makes it vague. The phrasing of "fatty acid carbon isotopes" won't be valued by some in the isotope community as it can sound a little bit colloquial. I would suggest changing to "d13C of fatty acids trace paleoproductivity off the coast of Adelie Land, Antarctica" or something along these lines.

**We agree with the reviewer that our approach may not be applicable as a productivity proxy in many other sites in Antarctica. However, the principal could be applied to other highly productive polynya environments on the Antarctic margins. We agree the title may therefore be slightly misleading so we will change it to: 'Exploring the use of compound-specific carbon isotopes as a palaeoproductivity proxy off the coast of Adélie Land, East Antarctica.'**

2. The manuscript gives a lot of space for trying to pin down a single, or majority, producer, for fatty acids such as C18. I think this is impossible as so many organisms produce C18 FA, and thus this discussion can be shortened and focused.

**Understanding the source(s) of the organic compounds is key to interpreting the signals recorded by $\delta^{13}C$. We do acknowledge in the manuscript that the $C_{18}$ fatty acid is likely to be produced by various different organisms and is unlikely to have a single producer. But even if the source cannot be pinned down to a specific species, understanding the predominant type of producer can make difference to the interpretation of the signal, for example a phytoplankton producer versus a higher trophic level source. However, there is evidence in the literature that the predominant producer of the $C_{18}$, within the context of an Antarctic polynya, can be narrowed down to a particular species i.e. *Phaeocystis antarctica*. Therefore, we feel it is important to include a discussion around what the predominant producer(s) are and the limitations of this. However, we are happy to reconsider our phrasing and condense this section.**

3. The changes observed in d13C are very small and some comments on how significant changes of 1‰o really are would be useful.

**The fatty acid $\delta^{13}$C data is discussed in the manuscript in comparison with other environmental $\delta^{13}$C signals to help understand the importance of the ~5‰ range in fatty acid $\delta^{13}$C. For example, we discuss previous studies which show the range of DIC $\delta^{13}$C in different water masses around Antarctica to be ~1.5‰ (lines 336 – 341) and the change in phytoplankton $\delta^{13}C_{org}$ due to anthropogenic $CO_2$ estimated to be up to 3.3‰ (lines 411 – 416). More importantly we discuss previous studies from the Ross Sea polynya, where sedimentary sterol $\delta^{13}$C has been shown to vary spatially by 5.6‰, from an area of high productivity within the polynya to an area low productivity further offshore. These changes follow a spatial variation in surface water $CO_2$ of <150 ppm to >400 ppm (lines 467-477). While our changes in fatty acid $\delta^{13}$C are not able to give a quantitative estimate of surface water $CO_2$ changes, the range in values is very consistent with the spatial variation observed in the Ross Sea sedimentary sterols suggesting they may reflect similar changes in $CO_2$ drawdown.**

**Since our fatty acid $\delta^{13}$C has a relatively high signal to noise ratio, we tend to limit our discussion to only large shifts in the data, greater than 1‰.**

**Furthermore, we have calculated our error on $\delta^{13}$C measurements as 0.26‰, based on duplicate analyses, suggesting that a shift of 1‰ is significant as an environmental signal.**

4. I can see a number of analytical issues that should be addressed. First of all, there is no explanation on how the correction for the methyl-group 13C values was carried out. This needs to be explained, or, if the C used for methylation has not been analysed for 13C and is not available anymore, and it is thus impossible to make this correction, it needs to be clearly acknowledged that values are not absolute.

**We thank the reviewer for pointing out that this was missing from our methods section. The $\delta^{13}$C was corrected for the extra C added during derivatization and in our final submission we will include some additional text in the methods about how this was done.**

The second issue is that the standard used (C19) is not the best for FAME as it is an n-alkane, and was only added post-extraction, hence analysis is semi-quantitative at best which needs to made clearer.

**We chose to use an *n*-alkane standard since these were not present in the FAME fraction and would therefore not risk co-eluting with any compounds within the sample. In our final submission we will make it clear that this was added post extraction and that our estimates of fatty acid concentration are therefore only semi-quantitative. This does not affect our $\delta^{13}$C measurements as these were corrected using an external Indiana F8 standard.**

5. Throughout the manuscript, often words such as "extremely", "very high", etc. are used – I would recommend a thorough edit removing these descriptions and replacing them with actual values that allow the reader to put them into context.

Line 68: Give a number instead of "extremely high" – how high?
Line 70: "highly productive" as above
**These comments refer to general descriptions of the polynya environments. However, in our final submission, we will add reference to Arrigo et al. (2015) which quantifies the annual net primary**

**production of the Dumont D'Urville polynya as 30.3 g C m$^{-2}$ a$^{-1}$ and the Mertz polynya as 39.9 g C m$^{-2}$ a$^{-1}$.**

Line 94: See comment 4 on internal standard – when was it added?
Does it really allow quantification at this point?

**As above, in our final submission we will make it clear that this was added post extraction and that our estimates of fatty acid concentration are therefore only semi-quantitative.**

Line 97: Are these values corrected for Me? Are these errors subsequently appropriately propagated? What is the significance of a change of just above 3 x SD (0.26 vs 1 ‰)?

**The $\delta^{13}$C errors are based on the duplicate measurements which we believe is a conservative approach to estimating error.**
**We refer to our response to point 3 above in which we discuss the significance of a change of 1‰**

Line 102: Which internal standards?

**To measure the HBI concentrations, we added 7 hexyl nonadecane (m/z 266) as an internal standard during the first extraction steps, following the Belt et al (2007) and Massé et al. (2011) protocols. We will include these details in the methods for our final submission.**

Line 194: Saying that a marine source is "entirely possible" sounds strange – do you want to say likely?

**Yes, in our final submission we will change this to likely.**

Lines 213-214: There are more novel studies on FA, Wakeham and also Hilary Close

**It is not clear which specific papers the reviewer is referring to here, or whether they are more relevant/add much to the discussion compared to the references already cited.**

Line 291: What do you mean by weaker coherence?

**What we mean is that there is less similarity between the C24 fatty acid and HBI triene concentration, compared with the C18 and HBI triene. In our final submission we will change the wording to make this clearer.**

Lines 547-549:
We know that there are many algae that make these FA so this is not likely to be resolved. At the same time, the non-distinctive nature of these molecules will make it difficult to apply this proxy to other settings where there are likely other producers. The whole paragraph is not particularly relevant and I would shorten and/or delete or move up so the work does not finish on a weak statement.

**We thank the reviewer for this helpful suggestion. In our final submission we will condense the last paragraph and move it up, and instead end with second paragraph so as not to finish the paper on the limitations of the proxy.**

---

## Author Comment (AC2) · 2 Jul 2020

**Response to Referee #2 (our replied are in bold)**

The high latitude region of the Southern Hemisphere which include Antarctic ice sheet and Southern Ocean is thought to play an important role in climate system, especially in long-climate change. Hence, it is important to investigate paleoclimate change the region to better understand Earth's climate. However, due to limited application of environmental proxies in the region, significant portions of Earth history, environmental records in the high latitude region are less developed than that of low and mid latitudes. Lower and higher molecular fatty acids that are produced by varieties of organisms in the ocean environment are ubiquitous in ocean sediments. Thus, fatty acids may have a potential as paleoenvironmental proxy. This study explores paleoclimatic utility of fatty acids in Southern Ocean sediments and suggests that stable carbon isotope ratio of the low (C18) and mid (C24) chain fatty acids could be used as productivity proxy in the sea ice area. Although further studies are needed to confirm robustness of the proxy, this study contributes development of biogeochemical proxy which has a potential to apply to high latitude ocean sediment. Hence, this study fits scope of Biogeosciences and suitable for publication in the journal. I have some comments on the article as below.

1. I would suggest to include some explanations that application of biomarker proxy is limited in polar regions into the introduction section (e.g. a powerful proxy such alkenone is not applicable in this region. HBI compounds, that are useful proxy of sea ice, are labile and cannot be applied to geological deep past. On the other hand, fatty acids are ubiquitous and abundantly detected even in old sediment and has a potential but its utility has not been investigated well). Such explanations highlight importance of this work.

**We thank the reviewer for this excellent suggestion and in our final submission will include an additional few sentences in the introduction section highlighting the utility of fatty acids as a proxy in this region.**

2. Although a number of fatty acids including C16 to C26 were abundantly detected in the studied samples (Figure S2), the authors show and discuss d13C results of C18 and C24 fatty acids only. I wonder why the authors focus the two compounds only. I suppose that aim of this paper is to investigate paleoclimatic utility of fatty acids in marine sediments. Hence, it is worth to also include results of the other compounds into the manuscript. I think many people are interested in results of other compounds and know how d13C profiles of other compounds look like. Including this significantly contributes to develop application of fatty acids in marine sediments to paleoclimate study.

**The C18 and C24 $\delta^{13}$C results were discussed in the paper as a representative of the short- and long-chain fatty acids groups respectively, as they appear to have different producers and thus offer a different signal. Both compounds yielded the highest quality isotope measurements whereas most other compounds had lower concentrations and therefore isotope measurements were of a poorer quality and not appropriate for publication. While we did obtain some isotope data on the C16 fatty acid, the peaks were not as clean (i.e. there was some coelution) and the data had higher error levels meaning we did not view the data as being as trustworthy or suitable for publication compared to the C18 and C24. We will, however, provide a spreadsheet of the concentration (all fatty acids) and $\delta^{13}$C (C18 and C24 fatty acids) data as part of the supplement. We refer the reviewer to**

**In lines 148 – 152 we do state the reason for focusing on these two compounds as follows: "The C18 and C24 FAs are the most abundant compounds within the SCFA and LCFA groups, respectively, and also the least correlated with each other both in the whole core (R2 = 0.5) and below 25 cm (R2 = 0.07), which suggests they are the most likely to be produced by different**

**organisms. Furthermore, these two compounds yielded the highest quality isotope measurements, due to their greater concentrations, clean baseline and minimal coeluting peaks (Fig. S2). Thus, these two compounds (C18 and C24) will be the focus of analysis and discussion."**

3. As for pCO2 effect on plankton d13C, important literature is missing in the manuscript (Pop et al., 1999, vol 13, 827-843, GBC). They measured d13C of POC along the north-south transect of the Southern Ocean and show significant negative correlation between dissolved CO2 and d13C of POC, suggesting strong control of pCO2 on d13C of phytoplankton. There needs to take into consideration the result for discussion.

**We thank the reviewer for pointing out this important study. In our final submission we will include discussion of Popp et al., 1999 to Section 4.2. Their findings are consistent with our data in which CO$_2$ plays a key role in driving carbon isotope compositions of marine sterol biomarkers.**

4. 4.2.3. Productivity section: The authors argue that changes in productivity is the most plausible driver for variability of fatty acid d13C recorded in the sediment core based on the results of previous studies conducted in the Southern Ocean (Villinski et al., 2008; Arrigo et al., 2015; Zhang et al., 2014). I basically agree that significant increase in productivity results in remarkable higher values of phytoplankton d13C in the polynya environment. However, those papers (Villinski et al., 2008; Arrigo et al., 2015; Zhang et al., 2014) all argue that observed increases in productivity in the regions are caused by meltwater input which promote surface stratification in summer time with reducing vertical mixing and supplying Fe, providing ideal condition for algal growth. Shadwick et al., GRL (2013) and Jack Pan et al., PlosOne (2019) also clearly show a significant correlation between meltwater fraction, chlorophyll concentration and surface water pCO2 drawdown. Especially, Shadwick et al., GRL (2013) investigates glacial meltwater impact on biological carbon drawdown in the studied region. Indeed, those paper shows lowering surface pCO2 happened in the regions where meltwater plume intruded. Regardless of sea ice fluctuations, plankton production takes place in summer when ice sheet melts. This suggests variability of meltwater input rather significantly affects productivity. Therefore, I would suggest to consider possible link between meltwater and productivity in the manuscript. Indeed, the observed resent increase in d13C of C16 fatty acid in sediment core is consistent with the fact of significant melting of Antarctic ice sheet for the past decades.

**In our final submission, we will add in a few sentences in the introduction section discussing what is known about the drivers of productivity in this region. Various papers have shown that _Phaeocystis antarctica_ for instance seems to predominate in more deeply mixed waters (e.g. Arrigo et al., 1999 and several other papers), as opposed to surface stratification being the main factor increasing primary productivity, even if meltwater and nutrient (e.g. Fe) transport plays a role on their production.**

**However, we feel that to include an interpretation of the drivers of productivity in our record is beyond the scope of this paper. We are working on a follow up paper which draws on other proxy data in which we can make more interpretations about the role of environmental factors such as sea ice and meltwater in driving productivity.**

5. F. cylindrus% and F. rhombica% records are shown in Figures 6 and 7, but the authors do not mention anything about those records in the manuscript. I wonder why those data are shown in the figures.

**We thank the reviewer for pointing out this oversight. We included relative abundances of these two diatoms in Fig. 6 (but not in Fig. 7 as the reviewer suggests) along with F. _kerguelensis_, F. _curta_**

and CRS as representatives of the main diatom groups, to show that shifts in fatty acid $\delta^{13}$C are most strongly covariant with *F. kerguensis* and not these other groups. We will add a few sentences explaining the reason for including these diatoms in our final submission.

---

## Referee Comment (RC3) · Anonymous Referee #1 · 3 Jul 2020

The authors suggest that "changes will be made" - does that mean they have been made in the resubmission? Crucially, the response does not detail where changes have been made. This should be shown with either indicative line numbers or reference to a tracked changes manuscript. Please resubmit this response with the necessary detail so the resubmitted MS can be evaluated accordingly as this is currently not possible.

---

## Referee Comment (RC4) · Anonymous Referee #3 · 3 Jul 2020

The manuscript by Ashley et al. proposes the use of the carbon isotope composition of selected fatty acids present in sediments as a palaeoproductivity proxy in an Antarctic polynya environment (Adélie Land). The topic, totally in line with the journal Biogeosciences, is worth being investigated as proxys of paleoproductivity, especially in Polar Regions, are still lacking. The authors present an interesting set of quantitative and isotopic data, and based on their expertise in polar environments, discuss their possible significance in terms of biogeochemical changes recorded in sediments. The approach is interesting but the discussion and the conclusions raised by the authors may appear a little over-optimistic as many assumptions are made and some potential biases are discarded too easily and/or overlooked. There are a number of issues that the authors should take into consideration before the manuscript can be considered for

publication. Comments are made chronologically, regardless of their importance.

Line 89 and manuscript throughout: It should be made clear in the manuscript that the data are based solely on free FA which represent only part of the total FA present in sediments (especially in modern to sub-recent sediments). If the selected FA indeed represent tracers of primary production, than it would be worth having a look at the bound (esterified) FA as well.

Lines 90-91: Please give more details on the use of BF3/MeOH as this treatment is known to be deleterious for some (poly)unsaturated FA.

Lines 91-94 and Fig. S2. Please give more detail on the chromatographic conditions used (for both GC and GC-MS analyses) and refer to figure S2. Also, the quality of the GC trace shown in figure S2 must be improved as, at such, a clear absence of unsaturated FA (which elute very close to saturated FA) is difficult to admit. As the authors know, the quality of compound specific 13C analyses is highly dependent on the purity of the compounds investigated and the absence of co-elution. Unsaturated FA often exhibit distinct 13C compositions compared to their saturated counterparts, so even small co-elution may significantly bias ïĄđ'13C values of saturated FA. An additional purification step using Si/Ag+ column chromatography may have been worth being investigated. Lastly, the peak attributed to the internal standard (C19 alkane) in Fig S2 is in fact most probably the C14 FA as it is not possible that the C19 FA elutes 15 minutes later than the C19 alkane. Please check peaks assignment (including the IS).

Lines 94-97. In line with the previous comment, more detail is undoubtedly required concerning CSIA. Which type of GC and conditions were used including the characteristics of the capillary column, the temperature of the interface and the oven, etc? Does 'Duplicate measures' means that each sample was analyzed twice? If so, the error given is a min-max and not a standard deviation. Were the measured ïĄđ'13C values corrected 1) for the methyl group added through derivatization and 2) for instrument

deviation using a standard mixture? Are the stable isotope ratios expressed relative to the standard Vienna Pee Dee Belemnite (V-PDB)?

Line 102: which IS were used for HBI?

Line 106: This is unclear as it sounds like a repetition of the previous sentence.

Lines 125-126: The sole presence of saturated FA in (sub)actual sediments of (hyper)productive areas is very unusual (this is an additional reason why a very clear GC trace is needed in Fig. S2 which could even be included in the main manuscript). Would it be possible that unsaturated FA were (partly) destroyed by the BF3 treatment?

Line 132: The actual figure 4 should become figure 2 and, consequently, actual figures 2 and 3 should become figures 3 and 4, respectively.

Actual figure 4: The upward displacement of either one or two GC trace(s) within each group would make the different trends easier to compare. The horizontal axis could also be homogenized with that of figures 5 and 6 (age or eventually both depth and age, and from right to left).

Line 144 and all along the manuscript: Please also give an estimated time span when speaking in cm depth.

Lines 166-168: In Dalsgaard et al., the mean proportion of C18:0 FA in Prymnesiophyceae is only 3%! Please specify it.

Line 170: 'higher preservation rate' may be misleading; replace with 'higher potential of preservation'.

Line 170: replace 'its proportion' with 'its relative proportion'.

Lines 166-181: This whole section deals with proportions of C18 FA in laboratory cultures which can show great differences with the environment. Could authors comment on this?

Lines 181-183 and more generally: This is one of my main concerns. The C18 FA can be produced by various type of (micro)organisms and assigning a single origin to this compound is rather daring. Authors should definitely support their hypothesis and comment about other potential sources of this compound such as bacteria, macrofauna, zooplankton, atmospheric inputs, land plants . . . One would also expect concentration profiles to be combined with d13C values to strengthen interpretations on the origin of individual biomarkers.

Line 184 and thereafter: The same comment (as that made for the C18 FA) holds for the C24 FA. In this case isotopic data could be additionally used to support a planktonic (vs terrestrial plants) origin.

Line 200 and thereafter: This is true but the degradation rates of lipid biomarkers appear strongly dependent on the redox conditions. Could authors give information on the redox state at the water-sediment interface and the possible influence of bioturbation in the surficial sediments?

Lines 227-229: Could this be due to an impact of bioturbation and/or to microbial production within the sediment?

Lines 257-259: A similarity between the concentration profiles of C18 FA, HBI triene and HBI diene is not obvious in figure 5. Authors are encouraged to reconsider/specify those words.

Lines 261-262: This sentence is not clear. Do authors mean: '.... and to diagenetic transformation within the sediments including sulfurisation (ref), isomerisation (ref) and cyclisation (ref) reactions'?

Lines 262-264: This statement is misleading and in contradiction with section 3.4. Clearly, one cannot speak about a better preservation in the top sediments. The concentrations of HBI reflect the flux of lipids reaching the seafloor while the decrease in concentration downcore reflects enhanced degradation in the first cm of sediments (yet

possibly including variations in productivity).

Lines 272-273: I agree but this holds true if diagenetic conditions remain the same through time. Any indication on potential variations in the redox state of the water column and water-sediment interface back in time?

Line 290: Again the concept that preservation of organic matter is better in surficial (younger) sediments is unfounded and in contradiction with section 3.4. It should be revised throughout the whole manuscript.

Lines 311-314 and thereafter: I don't think such a difference can be considered really significant (keeping in mind that the reproducibility was +/- 0.26 per mill). This statement might be a little far-fetched and I would suggest to remove it.

Lines 355-356. I am not convinced by this statement when looking at the d13C profile of the C24 FA which shows a clear trend towards lighter values (2-3 per mill) within the last 150 years. Authors are mentioning this trend later on (lines 375-376). Could this be linked to increased land plant inputs due to ice retreat?

Lines 366-367: please temper with 'do not tend to produce high proportions of this compound'.

Lines 372-378: This again is somewhat speculative. If both FA have distinct origins, than the diagenetic impact on their 13C composition may be significantly different. What about the possibility that either one or both FA are being produced in the top sediments?

Section 4.3 (lines 522-530) and conclusions (lines 540-541): As considered for FA in actual figures 2 and 3, a correlation table would help in highlighting putative relationships between lipid biomarker (concentration or d13C) profiles and specific phytoplanktonic species.

---

## Author Comment (AC3) · 20 Sep 2020

**Response to Referee #3 (our replies are in bold)**

The manuscript by Ashley et al. proposes the use of the carbon isotope composition of selected fatty acids present in sediments as a palaeoproductivity proxy in an Antarctic polynya environment (Adélie Land). The topic, totally in line with the journal Biogeosciences, is worth being investigated as proxys of paleoproductivity, especially in Polar Regions, are still lacking. The authors present an interesting set of quantitative and isotopic data, and based on their expertise in polar environments, discuss their possible significance in terms of biogeochemical changes recorded in sediments. The approach is interesting but the discussion and the conclusions raised by the authors may appear a little over-optimistic as many assumptions are made and some potential biases are discarded too easily and/or overlooked. There are a number of issues that the authors should take into consideration before the manuscript can be considered for publication. Comments are made chronologically, regardless of their importance.

Line 89 and manuscript throughout: It should be made clear in the manuscript that the data are based solely on free FA which represent only part of the total FA present in sediments (especially in modern to sub-recent sediments). If the selected FA indeed represent tracers of primary production, than it would be worth having a look at the bound (esterified) FA as well.

**We thank the reviewer for pointing this out. The focus of this paper is on free, saturated fatty acids, which is normal practice within palaeoceanography (due to their better preservation). In our final submission we will make it clearer.**

Lines 90-91: Please give more details on the use of BF3/MeOH as this treatment is known to be deleterious for some (poly)unsaturated FA.

**In our final submission we will add some more details about the use of BF3-MeOH. It is true that Morrisson (1964, p.605) and Klopfenstein (1971) seem to suggest that high concentrations (around ~50%) of BF3 can lead to some loss of unsaturated FAs. However, since we used a concentration of 14%, this should not be an issue for our samples. Furthermore, as the focus of this paper is on saturated fatty acids, a loss of polyunsaturated fatty acids would not affect our data.**

Lines 91-94 and Fig. S2. Please give more detail on the chromatographic conditions used (for both GC and GC-MS analyses) and refer to figure S2.

**We thank the reviewer for pointing out this omission. In our final submission we will add details of the chromatographic conditions used for GC-FID and GC-IRMS analysis, including the GC column dimensions, carrier gas and oven temperature programme.**

Also, the quality of the GC trace shown in figure S2 must be improved as, at such, a clear absence of unsaturated FA (which elute very close to saturated FA) is difficult to admit. As the authors know, the quality of compound specific 13C analyses is highly dependent on the purity of the compounds investigated and the absence of co-elution. Unsaturated FA often exhibit distinct 13C compositions compared to their saturated counterparts, so even small co-elution may significantly bias ïA̦d'13C values of saturated FA. An additional purification step using Si/Ag+ column chromatography may have been worth being investigated.

**It is not too clear what the reviewer expects in terms of a better GC trace. Any unsaturated FAs were below the detection limit of the GC and thus did not show up in any GC traces, hence their absence in figure S2. We agree that any coelution of unsaturated compounds would affect the d13C values, but we carefully checked the baseline of samples during analysis and can confirm**

**that any coelution of other peaks was minimal. In our final submission we will include the following GC trace which may be slightly clearer.**

[Figure]

Lastly, the peak attributed to the internal standard (C19 alkane) in Fig S2 is in fact most probably the C14 FA as it is not possible that the C19 FA elutes 15 minutes later than the C19 alkane. Please check peaks assignment (including the IS).

**We are quite confused as to why the reviewer believes our peak assignment is wrong, without even knowing the chromatographic conditions used (which they rightly point out in their previous comment). It certainly is possible that the C19 FA could elute 15 minutes later than the C19 alkane on a slow oven programme. The C19 alkane was prepared in a hexane solution and analysed on the GC to determine its retention time prior to its addition into the FAME samples, giving us confidence in the position of the internal standard. Furthermore, a selection of samples was analysed before and after addition of the C19 standard which also made it clear which peak was the C19. GC-MS analysis of a selection of samples indicated that the C14 fatty acid was either very weak or absent. In our final submission, we will include the GC trace noted above which shows stronger peaks and here a very small C14 fatty acid peak is visible (at 13.9 minutes) next to the C19 alkane (at 14.3 minutes) indicating that they are in fact different peaks.**

Lines 94-97. In line with the previous comment, more detail is undoubtedly required concerning CSIA. Which type of GC and conditions were used including the characteristics of the capillary column, the temperature of the interface and the oven, etc?

**In our final submission we will add details of the chromatographic conditions used for GC-FID and GC-IRMS analysis, including the GC column dimensions, carrier gas and oven temperature programme.**

Does 'Duplicate measures' means that each sample was analyzed twice?

**Yes**

If so, the error given is a min-max and not a standard deviation.

**We will change the wording and remove the use of standard deviation here.**

Were the measured ïA¸d'13C values corrected 1) for the methyl group added through derivatization

**Yes the carbon isotopes were corrected for the addition of the methyl group due to derivatization and we will add details of this correction in our final submission.**

and 2) for instrument deviation using a standard mixture?

**No, the d13C values were not corrected for the instrument deviation, but this was monitored throughout analysis using external standards (F8, Indiana) and remained low throughout.**

Are the stable isotope ratios expressed relative to the standard Vienna Pee Dee Belemnite (V-PDB)?

**Yes, they are expressed relative to VPDB. We will include mention of this in our final submission.**

Line 102: which IS were used for HBI?

**To measure the HBI concentrations, we added 7 hexyl nonadecane (m/z 266) as an internal standard during the first extraction steps, following the Belt et al (2007) and Massé et al. (2011) protocols. We will include these details in the methods for our final submission.**

Line 106: This is unclear as it sounds like a repetition of the previous sentence.

**We will refine these sentences in our final submission.**

Lines 125-126: The sole presence of saturated FA in (sub)actual sediments of (hyper) productive areas is very unusual (this is an additional reason why a very clear GC trace is needed in Fig. S2 which could even be included in the main manuscript). Would it be possible that unsaturated FA were (partly) destroyed by the BF3 treatment?

**See lines 211-233. Here we discuss the breakdown of unsaturated and short-chain fatty acids in the water column and on the sea floor before burial. This most likely explains the sole presence of saturated FAs in our samples (Haddad et al., 1992; Matsuda, 1978; Colombo et al., 1997). Our understanding is that this is not unusual, unless the reviewer would like to share any specific references on this. The hyper-productive environment offshore Adélie Land is unique and not well studied thus it is hard to know how it compares to other sites. Although further investigations are certainly needed, we think it is unlikely that unsaturated fatty acids would be destroyed by BF3 due to the low concentration we used (see response to previous comment on this).**

Line 132: The actual figure 4 should become figure 2 and, consequently, actual figures 2 and 3 should become figures 3 and 4, respectively.

**We will change the order of these figures in our final submission.**

Actual figure 4: The upward displacement of either one or two GC trace(s) within each group would make the different trends easier to compare. The horizontal axis could also be homogenized with that of figures 5 and 6 (age or eventually both depth and age, and from right to left).

**The overlap of the FAME concentration plots shows the strong coherence between the datasets, which wouldn't be as clear if they were offset, thus we prefer to keep it this way. We choose to plot this data against depth in this figure since this section is dealing with how the FAME concentrations change downcore and how the different compounds compare to each other and age is not particularly relevant until later in the discussion. It is not really possible to have both age and depth on the x-axis since the age model is not completely linear. We will provide the data in the supplement so readers will be able to look at both depth and age if they wish.**

Line 144 and all along the manuscript: Please also give an estimated time span when speaking in cm depth.

**We will add an estimated time span in our final submission.**

Lines 166-168: In Dalsgaard et al., the mean proportion of C18:0 FA in Prymnesiophyceae is only 3%! Please specify it.

**This is correct. We pointed out in lines 168-170 that, in this study, the majority of FAs produced were the unsaturated form which are preferentially broken down in the water column and sediments (Haddad et al., 1992; Matsuda, 1978; Colombo et al., 1997). Thus, although the C18 FA represents only 3% of the *total* FA fraction, its higher preservation rate compared to unsaturated fatty acids, increases its proportion in the sediment.**

Line 170: 'higher preservation rate' may be misleading; replace with 'higher potential of preservation'.
**We will change the wording accordingly in our final submission.**

Line 170: replace 'its proportion' with 'its relative proportion'.
**We will change the wording accordingly in our final submission.**

Lines 166-181: This whole section deals with proportions of C18 FA in laboratory cultures which can show great differences with the environment. Could authors comment on this?

**Since there are very few studies looking into the different algal producers of C18 FA it is tentative to comment on potential differences between the results of in vitro and in situ studies. In the current knowledge stage, laboratory experiments are essential to document which algae, and under which conditions, synthesize the C18 FA. We will add few words in our final submission on the fact that this finding is based on laboratory cultures and on the potential limitations.**

Lines 181-183 and more generally: This is one of my main concerns. The C18 FA can be produced by various type of (micro)organisms and assigning a single origin to this compound is rather daring. Authors should definitely support their hypothesis and comment about other potential sources of this compound such as bacteria, macrofauna, zooplankton, atmospheric inputs, land plants… One would also expect concentration profiles to be combined with d13C values to strengthen interpretations on the origin of individual biomarkers.

**See lines 308-316. Here we include the d13C values in our interpretation of the source of fatty acids which supports a pelagic phytoplankton source. Our suggestion of *Phaeocystis antarctica* as the main producer of C18 is clearly presented as the most likely dominant source based on the available information and is a conservative suggestion. We point out that contributions from other sources such as diatoms or dinoflagellates cannot be excluded. Inputs from land plants and atmospheric inputs are highly unlikely due to the location of the core (Antarctica) and the highly productive nature of the water column.**

Line 184 and thereafter: The same comment (as that made for the C18 FA) holds for the C24 FA. In this case isotopic data could be additionally used to support a planktonic (vs terrestrial plants) origin.

**We are very cautious in our interpretation of the C24 fatty acid and do not assign a specific source. As we point out, contributions from terrestrial plants are highly unlikely due to the lack of land plants proximal to the core and the highly productive nature of the water column in this area.**

Line 200 and thereafter: This is true but the degradation rates of lipid biomarkers appear strongly dependent on the redox conditions. Could authors give information on the redox state at the water-sediment interface and the possible influence of bioturbation in the surficial sediments?

**The preservation of annual to sub-annual laminae throughout the core indicates very reduced bioturbation and the presence of dysoxic to anoxic bottom waters. However, we argue that much of the degradation takes place within the water column which is well-mixed and oxygenated, as well as in the surface sediments. This is a highly productive environment involving many trophic levels thus recycling of material in the water column will be substantial resulting in anoxic bottom waters. We don't have information on the redox conditions, it has never been undertaken and this would be very difficult to monitor at such a remote and hostile location.**

Lines 227-229: Could this be due to an impact of bioturbation and/or to microbial production within the sediment?

**This is highly unlikely due to lack of bioturbation and anoxic bottom waters. While we cannot rule out anaerobic microbial production in the surface sediments, this appears to be unlikely due to the consistent profile of FA homologues. If there was a major contribution from in situ microbes, we would expect a change in the FA profile such as the presence of branched fatty acids etc. in younger samples.**

Lines 257-259: A similarity between the concentration profiles of C18 FA, HBI triene and HBI diene is not obvious in figure 5. Authors are encouraged to reconsider/specify those words.

**In lines 257-258 we state that "one key similarity between both the HBI diene and triene, and the FA concentrations is that the highest concentrations are found in the youngest sediments." Figure 5 is split into two sections – the 1550-1950 period and 1950-2000 which have different y-axes. The y-axes for the 1950-2000 period (shown on the right) have much higher values for all four plots than the older period (shown on the left) since the concentrations in this period are much higher. Plotting the whole record on the same y-axis would mean that the plot is dominated by the high concentration in the top part of the core and the smaller-scale changes would not be visible, hence choosing to split it up. Thus, the similarity between the fatty acids and HBIs in having higher concentrations in the top part of the core is clear from the higher values in the y-axes on the right-hand side of the figure. The higher concentrations of fatty acids in the top of core are clearly shown in Figure 4.**

Lines 261-262: This sentence is not clear. Do authors mean: '.... and to diagenetic transformation within the sediments including sulfurisation (ref), isomerisation (ref) and cyclisation (ref) reactions'

**We will amend the sentence in our final submission to: "Concentrations of HBIs are also susceptible to degradation through the water column through visible light induced photo-degradation (Belt and Müller, 2013) and diagenetic effects within the sediments including sulphurisation (Sinninghe Damsté et al., 2007), isomerisation and cyclisation (Belt et al., 2000)."**

Lines 262-264: This statement is misleading and in contradiction with section 3.4. Clearly, one cannot speak about a better preservation in the top sediments. The concentrations of HBI reflect the flux of lipids reaching the seafloor while the decrease in concentration downcore reflects enhanced degradation in the first cm of sediments (yet possibly including variations in productivity).

**When we say better preservation, we mean that younger sediments have been less affected by diagenetic effects since degradation increases over time, and that explains why we have higher concentrations in the top of the core. However, in our final submission we can change to the wording slightly to make it clearer. We agree with the reviewer that concentrations in the core will reflect both the flux of lipids reaching the seafloor and diagenetic effects. We discuss this in lines 265-273 where we suggest that changes in concentration will be affected by both changes in preservation processes and any change in production of compounds in the surface waters.**

Lines 272-273: I agree but this holds true if diagenetic conditions remain the same through time. Any indication on potential variations in the redox state of the water column and water-sediment interface back in time?

**Unfortunately, we do not have data on the redox state of the water column as it has never been undertaken. Mn is sometimes used as a proxy for redox conditions at the water-sediment interface (Jimenez-Espejo et al., 2019). Unfortunately, this element has not been measured in DTGC2011 core.**

Line 290: Again the concept that preservation of organic matter is better in surficial (younger) sediments is unfounded and in contradiction with section 3.4. It should be revised throughout the whole manuscript.

**As mentioned above, in our final submission we can change to the wording slightly to explain this more clearly.**

Lines 311-314 and thereafter: I don't think such a difference can be considered really significant (keeping in mind that the reproducibility was +/- 0.26 per mill). This statement might be a little far-fetched and I would suggest to remove it.

**We are not convinced that a difference of 0.5 per mil should be dismissed as insignificant, since it is twice the reproducibility level and this is based on the mean of all samples in the core. Besides, this is only a tentative suggestion and is certainly worth considering. We will present it as such in the revised version.**

Lines 355-356. I am not convinced by this statement when looking at the d13C profile of the C24 FA which shows a clear trend towards lighter values (2-3 per mill) within the last 150 years. Authors are mentioning this trend later on (lines 375-376). Could this be linked to increased land plant inputs due to ice retreat?

**Over the last 200 years, the C24 FA d13C values initially trend towards more positive values before becoming more negative in the last ~100 years. In contrast, the C18 FA shows no clear long-term trend over the past 200 years except for a relatively rapid shift towards more positive values after ~1950 C.E. If atmospheric CO2 was a key driver of fatty acid d13C, then we would expect both compounds to respond together, showing a trend towards more negative values over the last 200 years which neither of them do. However, we thank the reviewer for highlighting the trend in C24 FA and we will add an additional sentence to point this out.**

**We do not believe that the trend towards more negative values in the C24 FA d13C would be due to increased land plant inputs. It is true that parts of the Antarctica Peninsula have experienced an increase in mosses due to recent ice retreat, however, there is no evidence of recent ice retreat in Adélie Land and very few land plants are present in this area. Our core site is situated 50km off the**

**coast meaning that even if there were land plants in Adélie Land, their contribution would be very low.**

Lines 366-367: please temper with 'do not tend to produce high proportions of this compound'.

**We will amend the sentence as suggested.**

Lines 372-378: This again is somewhat speculative. If both FA have distinct origins, than the diagenetic impact on their 13C composition may be significantly different. What about the possibility that either one or both FA are being produced in the top sediments?

**We disagree that the C24 and C18 would have such different diagenetic pathways. While we cannot rule out this suggestion completely, we are not aware of any literature to suggest this would be the case. As mentioned before, there is no evidence of any bioturbation or metazoan benthic activity in the sediments, and the bottom waters are known to be anoxic meaning any fatty acid producers within the sediments would be very limited.**

Section 4.3 (lines 522-530) and conclusions (lines 540-541): As considered for FA in actual figures 2 and 3, a correlation table would help in highlighting putative relationships between lipid biomarker (concentration or d13C) profiles and specific phytoplanktonic species.

**While we agree that this could be helpful, unfortunately the biomarker and diatom data were taken from adjacent samples (thus have different depths and ages) meaning they cannot be directly compared (unless they were resampled which would introduce errors). In contrast, comparison of different fatty acid compounds was possible since they which were all present within each sample meaning they were analysed simultaneously. Furthermore, due to the nature of the data, having high frequency-high amplitude changes due to dynamic environment, and the fact that they are different types of data, we do not think that a correlation table comparing diatoms and biomarkers would be useful. For this reason, it is not common in palaeoclimatic to look at correlations and is generally considered more useful to look at broad coherence between datasets which may change downcore. Therefore, we think it is more useful to plot the downcore records together in order to see the coherence between them.**

---

## Author Comment (AC4) · 3 Oct 2020

We were simply asked to respond to the comments at this stage. It will be up to the editor to decide if we can submit a revised manuscript. Hopefully they have clarified this process to the reviewer....

---

## Author Response (AR1)

**Response to reviewer 1 (our replies are in bold)**

Ashley et al. present an assessment on the usefulness of d13C of fatty acids to assess paleoproductivity in an Antarctic coastal setting The manuscript is well-written, the data appropriate and extensive, and the research question interesting and relevant. The rationale for this work is fully explained, the introduction is clear and the methodology is sound. The main results and discussion section is generally clear, but not enough attention and focus is given to linking the data to productivity. At present, it almost looks like productivity was chosen because the trends could not be explained by anything else. I am sure this is not the case, but it needs to be made clearer for the reader as well.

**We have added an additional few sentences to the introduction explaining the starting point for this work and our d13C interpretations and our aims for the discussion (lines 60-66).**

There are a few criticisms I have which ought to be addressed before this manuscript is ready for publication.

1. The manuscript is focusing on one specific site, and while the observed links to productivity are observed here, the site is very particular and in no way is this ready to be extrapolated at all to any other sites in Antarctica or any other settings. Hence, the title is a little presumptuous, while at the same time the phrasing as a question makes it vague. The phrasing of "fatty acid carbon isotopes" won't be valued by some in the isotope community as it can sound a little bit colloquial. I would suggest changing to "d13C of fatty acids trace paleoproductivity off the coast of Adelie Land, Antarctica" or something along these lines.

**We have changed the title to: 'Exploring the use of compound-specific carbon isotopes as a palaeoproductivity proxy off the coast of Adélie Land, East Antarctica.'**

2. The manuscript gives a lot of space for trying to pin down a single, or majority, producer, for fatty acids such as C18. I think this is impossible as so many organisms produce C18 FA, and thus this discussion can be shortened and focused.

**We have shortened section 3.2.**

3. The changes observed in d13C are very small and some comments on how significant changes of 1‰o really are would be useful.

**As mentioned previously, the fatty acid $\delta^{13}$C data is discussed in the manuscript in comparison with other environmental $\delta^{13}$C signals to help understand the importance of the ~5‰ range in fatty acid $\delta^{13}$C, which we feel is sufficient to help the reader understand the significance of such changes.**

4. I can see a number of analytical issues that should be addressed. First of all, there is no explanation on how the correction for the methyl-group 13C values was carried out. This needs to be explained, or, if the C used for methylation has not been analysed for 13C and is not available anymore, and it is thus impossible to make this correction, it needs to be clearly acknowledged that values are not absolute.

**We have included details about how the d13C values were correction for the derivatization (lines 144-153)**

The second issue is that the standard used (C19) is not the best for FAME as it is an n-alkane, and was only added post-extraction, hence analysis is semi-quantitative at best which needs to made clearer.

**We had added some text at the start of Section 3.1. mentioning that the C19 alkane was added post-extraction and hence concentrations estimates are semi-quantitative (lines 188-190).**

5. Throughout the manuscript, often words such as "extremely", "very high", etc. are used – I would recommend a thorough edit removing these descriptions and replacing them with actual values that allow the reader to put them into context.

Line 68: Give a number instead of "extremely high" – how high?
Line 70: "highly productive" as above

**We have edited this to include specific annual net primary productivity rates (lines 85-87)**

Line 94: See comment 4 on internal standard – when was it added?
Does it really allow quantification at this point?

**As above, we have mentioned this at the start of section 3.1 (lines 188-190)**

Line 97: Are these values corrected for Me? Are these errors subsequently appropriately propagated? What is the significance of a change of just above 3 x SD (0.26 vs 1 ‰)?

**As mentioned previously, the $\delta^{13}$C errors are based on the duplicate measurements which we believe is a conservative approach to estimating error. We refer to our response to point 3 above in which we discuss the significance of a change of 1‰**

Line 102: Which internal standards?

**We have included details of the internal standard used in the methods section (lines 158-160)**

Line 194: Saying that a marine source is "entirely possible" sounds strange – do you want to say likely?

**We have changed this to likely (line 261)**

Lines 213-214: There are more novel studies on FA, Wakeham and also Hilary Close

**It is not clear which specific papers the reviewer is referring to here, or whether they are more relevant/add much to the discussion compared to the references already cited.**

Line 291: What do you mean by weaker coherence?

**We have changed the wording to: 'There is less similarity between the C24 and both the HBI triene also HBI diene, (compared to the coherence between C18 FA and HBI triene), which suggests that the C24 FA is predominantly produced by an organism which is not associated with sea ice, and thus instead with more open waters.' (lines 359-362)**

Lines 547-549:

We know that there are many algae that make these FA so this is not likely to be resolved. At the same time, the non-distinctive nature of these molecules will make it difficult to apply this proxy to other settings where there are likely other producers. The whole paragraph is not particularly relevant and I would shorten and/or delete or move up so the work does not finish on a weak statement.

**We have moved this paragraph further up in the conclusion section so it doesn't end the paper (lines 617-623).**

**Response to Referee #2 (our replied are in bold)**

The high latitude region of the Southern Hemisphere which include Antarctic ice sheet and Southern Ocean is thought to play an important role in climate system, especially in long-climate change. Hence, it is important to investigate paleoclimate change the region to better understand Earth's climate. However, due to limited application of environmental proxies in the region, significant portions of Earth history, environmental records in the high latitude region are less developed than that of low and mid latitudes. Lower and higher molecular fatty acids that are produced by varieties of organisms in the ocean environment are ubiquitous in ocean sediments. Thus, fatty acids may have a potential as paleoenvironmental proxy. This study explores paleoclimatic utility of fatty acids in Southern Ocean sediments and suggests that stable carbon isotope ratio of the low (C18) and mid (C24) chain fatty acids could be used as productivity proxy in the sea ice area. Although further studies are needed to confirm robustness of the proxy, this study contributes development of biogeochemical proxy which has a potential to apply to high latitude ocean sediment. Hence, this study fits scope of Biogeosciences and suitable for publication in the journal. I have some comments on the article as below.

1. I would suggest to include some explanations that application of biomarker proxy is limited in polar regions into the introduction section (e.g. a powerful proxy such alkenone is not applicable in this region. HBI compounds, that are useful proxy of sea ice, are labile and cannot be applied to geological deep past. On the other hand, fatty acids are ubiquitous and abundantly detected even in old sediment and has a potential but its utility has not been investigated well). Such explanations highlight importance of this work.

**We have included a few additional sentences in the introduction explaining the potential utility of fatty acids as a paleo proxy in this region (lines 54-59)**

2. Although a number of fatty acids including C16 to C26 were abundantly detected in the studied samples (Figure S2), the authors show and discuss d13C results of C18 and C24 fatty acids only. I wonder why the authors focus the two compounds only. I suppose that aim of this paper is to investigate paleoclimatic utility of fatty acids in marine sediments. Hence, it is worth to also include results of the other compounds into the manuscript. I think many people are interested in results of other compounds and know how d13C profiles of other compounds look like. Including this significantly contributes to develop application of fatty acids in marine sediments to paleoclimate study.

**As mentioned previously, we feel we have already answered this in the last paragraph of section 3.1. We will, however, provide a spreadsheet of the concentration (all fatty acids) and $\delta^{13}$C (C18 and C24 fatty acids) data as part of the supplement.**

3. As for pCO2 effect on plankton d13C, important literature is missing in the manuscript (Pop et al., 1999, vol 13, 827-843, GBC). They measured d13C of POC along the north-south transect of the Southern Ocean and show significant negative correlation between dissolved CO2 and d13C of POC, suggesting strong control of pCO2 on d13C of phytoplankton. There needs to take into consideration the result for discussion.

**We have included mention of the findings of Popp et al., 1999 to our discussion in Section 4.2 (lines 461-463).**

4. 4.2.3. Productivity section: The authors argue that changes in productivity is the most plausible driver for variability of fatty acid d13C recorded in the sediment core based on the results of previous studies conducted in the Southern Ocean (Villinski et al., 2008; Arrigo et al., 2015; Zhang et al., 2014). I basically agree that significant increase in productivity results in remarkable higher values of phytoplankton d13C in the polynya environment. However, those papers (Villinski et al., 2008; Arrigo et al., 2015; Zhang et al., 2014) all argue that observed increases in productivity in the regions are caused by meltwater input which promote surface stratification in summer time with reducing vertical mixing and supplying Fe, providing ideal condition for algal growth. Shadwick et al., GRL (2013) and Jack Pan et al., PlosOne (2019) also clearly show a significant correlation between meltwater fraction, chlorophyll concentration and surface water pCO2 drawdown. Especially, Shadwick et al., GRL (2013) investigates glacial meltwater impact on biological carbon drawdown in the studied region. Indeed, those paper shows lowering surface pCO2 happened in the regions where meltwater plume intruded. Regardless of sea ice fluctuations, plankton production takes place in summer when ice sheet melts. This suggests variability of meltwater input rather significantly affects productivity. Therefore, I would suggest to consider possible link between meltwater and productivity in the manuscript. Indeed, the observed resent increase in d13C of C16 fatty acid in sediment core is consistent with the fact of significant melting of Antarctic ice sheet for the past decades.

**As mentioned in our previous response, we feel that to include an interpretation of the drivers of productivity in our record is beyond the scope of this paper. However, we have added a few addition lines into the environmental setting section mentioned various drivers of productivity in the region (lines 87-92).**

5. F. cylindrus% and F. rhombica% records are shown in Figures 6 and 7, but the authors do not mention anything about those records in the manuscript. I wonder why those data are shown in the figures.

**We included relative abundances of these two diatoms in Fig. 6 along with F. *kerguelensis*, F. *curta* and CRS as representatives of the main diatom groups. We have included mention of the different diatom species included in Fig. 4 (lines 575-577) and an additional sentence pointing out the lack of similarity with F cylindrus and F. rhombica (lines 610-611).**

**Response to Referee #3 (our replies are in bold)**

The manuscript by Ashley et al. proposes the use of the carbon isotope composition of selected fatty acids present in sediments as a palaeoproductivity proxy in an Antarctic polynya environment (Adélie Land). The topic, totally in line with the journal Biogeosciences, is worth being investigated as proxys of paleoproductivity, especially in Polar Regions, are still lacking. The authors present an interesting set of quantitative and isotopic data, and based on their expertise in polar environments, discuss their possible significance in terms of biogeochemical changes recorded in sediments. The approach is interesting but the discussion and the conclusions raised by the authors may appear a little over-optimistic as many assumptions are made and some potential biases are discarded too easily and/or overlooked. There are a number of issues that the authors should take into consideration before the manuscript can be considered for publication. Comments are made chronologically, regardless of their importance.

Line 89 and manuscript throughout: It should be made clear in the manuscript that the data are based solely on free FA which represent only part of the total FA present in sediments (especially in modern to sub-recent sediments). If the selected FA indeed represent tracers of primary production, than it would be worth having a look at the bound (esterified) FA as well.

**We have made it clear in the introduction (and the start of section 3) that this paper is based on free, saturated fatty acids (line 52 and 183)**

Lines 90-91: Please give more details on the use of BF3/MeOH as this treatment is known to be deleterious for some (poly)unsaturated FA.

**We have included the concentration of BF3 used (line 111-112). As stated above, we have made clear that this paper is on saturated fatty acids, thus any effect of BF3 on polyunsaturated fatty acids should not affect our data.**

Lines 91-94 and Fig. S2. Please give more detail on the chromatographic conditions used (for both GC and GC-MS analyses) and refer to figure S2.

**We have now included details of the chromatographic conditions used for GC-FID, GC-MS and GC-IRMS analysis, including the GC column dimensions, carrier gas and oven temperature programme (lines 114-126).**

Also, the quality of the GC trace shown in figure S2 must be improved as, at such, a clear absence of unsaturated FA (which elute very close to saturated FA) is difficult to admit. As the authors know, the quality of compound specific 13C analyses is highly dependent on the purity of the compounds investigated and the absence of co-elution. Unsaturated FA often exhibit distinct 13C compositions compared to their saturated counterparts, so even small co-elution may significantly bias ïA¸d'13C values of saturated FA. An additional purification step using Si/Ag+ column chromatography may have been worth being investigated.

**As mentioned in our previous response, any unsaturated FAs were below the detection limit of the GC and thus did not show up in any GC traces, hence their absence in figure S2. We carefully checked the baseline of samples during analysis and can confirm that any coelution of other peaks was minimal. We have now included an additional GC trace (below) which we hope is slightly clearer (Fig S2b).**

[Figure]

Lastly, the peak attributed to the internal standard (C19 alkane) in Fig S2 is in fact most probably the C14 FA as it is not possible that the C19 FA elutes 15 minutes later than the C19 alkane. Please check peaks assignment (including the IS).

**We have now included the GC trace noted above (Fig S2b) in which a very small C14 fatty acid peak is visible (at 13.9 minutes) next to the C19 alkane (at 14.3 minutes) indicating that they are in fact different peaks.**

Lines 94-97. In line with the previous comment, more detail is undoubtedly required concerning CSIA. Which type of GC and conditions were used including the characteristics of the capillary column, the temperature of the interface and the oven, etc?

**We have now added details of the chromatographic conditions used for GC-IRMS analysis, including the GC column dimensions, carrier gas and oven temperature programme (lines 114-126).**

Does 'Duplicate measures' means that each sample was analyzed twice?

**Yes**

If so, the error given is a min-max and not a standard deviation.

**We have changed the wording to remove standard deviation (line 142)**

Were the measured ïA˛d'13C values corrected 1) for the methyl group added through derivatization

**We have added details of the correction for derivatization (lines 144-153)**

and 2) for instrument deviation using a standard mixture?

**No, the d13C values were not corrected for the instrument deviation, but this was monitored throughout analysis using external standards (F8, Indiana) and remained low throughout.**

Are the stable isotope ratios expressed relative to the standard Vienna Pee Dee Belemnite (V-PDB)?

**Yes, they are expressed relative to VPDB. We have included mention of this in the methods section (line 127-128)**

Line 102: which IS were used for HBI?

**We have included details of the internal standard used for the HBI measurement in the methods (line 158-159)**

Line 106: This is unclear as it sounds like a repetition of the previous sentence.

**We have re-worded this (line 163)**

Lines 125-126: The sole presence of saturated FA in (sub)actual sediments of (hyper) productive areas is very unusual (this is an additional reason why a very clear GC trace is needed in Fig. S2 which could even be included in the main manuscript). Would it be possible that unsaturated FA were (partly) destroyed by the BF3 treatment?

**As mentioned in our previous response, we refer to lines 211-233. Our understanding is that the sole presence of saturated FA is not unusual. The hyper-productive environment offshore Adélie Land is unique and not well studied thus it is hard to know how it compares to other sites. We believe it is unlikely that unsaturated fatty acids would be destroyed by BF3 due to the low concentration we used.**

Line 132: The actual figure 4 should become figure 2 and, consequently, actual figures 2 and 3 should become figures 3 and 4, respectively.

**We have now changed the order of these figures.**

Actual figure 4: The upward displacement of either one or two GC trace(s) within each group would make the different trends easier to compare. The horizontal axis could also be homogenized with that of figures 5 and 6 (age or eventually both depth and age, and from right to left).

**The overlap of the FAME concentration plots shows the strong coherence between the datasets, which wouldn't be as clear if they were offset, thus we prefer to keep it this way. We choose to plot this data against depth in this figure since this section is dealing with how the FAME concentrations change downcore and how the different compounds compare to each other and age is not particularly relevant until later in the discussion. It is not really possible to have both age and depth on the x-axis since the age model is not completely linear. We will provide the data in the supplement so readers will be able to look at both depth and age if they wish.**

Line 144 and all along the manuscript: Please also give an estimated time span when speaking in cm depth.

**We have included mention of the rough age of the sediments when referring only to depth (lines 205, 296 and 451)**

Lines 166-168: In Dalsgaard et al., the mean proportion of C18:0 FA in Prymnesiophyceae is only 3%! Please specify it.

**As mentioned in our previous response, we pointed out in lines 168-170 that the majority of FAs produced were the unsaturated form which are preferentially broken down in the water column and sediments (Haddad et al., 1992; Matsuda, 1978; Colombo et al., 1997). Thus, although the C18 FA represents only 3% of the *total* FA fraction, its higher preservation rate compared to unsaturated fatty acids, increases its proportion in the sediment.**

Line 170: 'higher preservation rate' may be misleading; replace with 'higher potential of preservation'.

Line 170: replace 'its proportion' with 'its relative proportion'.

**This sentence has since been removed.**

Lines 166-181: This whole section deals with proportions of C18 FA in laboratory cultures which can show great differences with the environment. Could authors comment on this?

**This paragraph has since been removed.**

Lines 181-183 and more generally: This is one of my main concerns. The C18 FA can be produced by various type of (micro)organisms and assigning a single origin to this compound is rather daring. Authors should definitely support their hypothesis and comment about other potential sources of this compound such as bacteria, macrofauna, zooplankton, atmospheric inputs, land plants... One would also expect concentration profiles to be combined with d13C values to strengthen interpretations on the origin of individual biomarkers.

**As mentioned previous, in the second paragraph of section 4 we include the d13C values in our interpretation of the source of fatty acids which supports a pelagic phytoplankton source. Our suggestion of *Phaeocystis antarctica* as the main producer of C18 is clearly presented as the most likely dominant source based on the available information and is a conservative suggestion. We point out that contributions from other sources such as diatoms or dinoflagellates cannot be excluded. Inputs from land plants and atmospheric inputs are highly unlikely due to the location of the core (Antarctica) and the highly productive nature of the water column.**

Line 184 and thereafter: The same comment (as that made for the C18 FA) holds for the C24 FA. In this case isotopic data could be additionally used to support a planktonic (vs terrestrial plants) origin.

**We are very cautious in our interpretation of the C24 fatty acid and do not assign a specific source. As we point out, contributions from terrestrial plants are highly unlikely due to the lack of land plants proximal to the core and the highly productive nature of the water column in this area.**

Line 200 and thereafter: This is true but the degradation rates of lipid biomarkers appear strongly dependent on the redox conditions. Could authors give information on the redox state at the water-sediment interface and the possible influence of bioturbation in the surficial sediments?

**The preservation of annual to sub-annual laminae throughout the core indicates very reduced bioturbation and the presence of dysoxic to anoxic bottom waters. However, we argue that much of the degradation takes place within the water column which is well-mixed and oxygenated, as well as in the surface sediments. This is a highly productive environment involving many trophic levels thus recycling of material in the water column will be substantial resulting in anoxic bottom waters. We don't have information on the redox conditions, it has never been undertaken and this would be very difficult to monitor at such a remote and hostile location.**

Lines 227-229: Could this be due to an impact of bioturbation and/or to microbial production within the sediment?

**This is highly unlikely due to lack of bioturbation and anoxic bottom waters. While we cannot rule out anaerobic microbial production in the surface sediments, this appears to be unlikely due to the consistent profile of FA homologues. If there was a major contribution from in situ microbes,**

**we would expect a change in the FA profile such as the presence of branched fatty acids etc. in younger samples.**

Lines 257-259: A similarity between the concentration profiles of C18 FA, HBI triene and HBI diene is not obvious in figure 5. Authors are encouraged to reconsider/specify those words.

**In lines 257-258 we state that "one key similarity between both the HBI diene and triene, and the FA concentrations is that the highest concentrations are found in the youngest sediments." Figure 5 is split into two sections – the 1550-1950 period and 1950-2000 which have different y-axes. The y-axes for the 1950-2000 period (shown on the right) have much higher values for all four plots than the older period (shown on the left) since the concentrations in this period are much higher. Plotting the whole record on the same y-axis would mean that the plot is dominated by the high concentration in the top part of the core and the smaller-scale changes would not be visible, hence choosing to split it up. Thus, the similarity between the fatty acids and HBIs in having higher concentrations in the top part of the core is clear from the higher values in the y-axes on the right-hand side of the figure. The higher concentrations of fatty acids in the top of core are clearly shown in Figure 4.**

Lines 261-262: This sentence is not clear. Do authors mean: '.... and to diagenetic transformation within the sediments including sulfurisation (ref), isomerisation (ref) and cyclisation (ref) reactions'

**We have amended this sentence (lines 327-329).**

Lines 262-264: This statement is misleading and in contradiction with section 3.4. Clearly, one cannot speak about a better preservation in the top sediments. The concentrations of HBI reflect the flux of lipids reaching the seafloor while the decrease in concentration downcore reflects enhanced degradation in the first cm of sediments (yet possibly including variations in productivity).

**We have amended this sentence to make it clearer what we mean (lines 330-331)**

Lines 272-273: I agree but this holds true if diagenetic conditions remain the same through time. Any indication on potential variations in the redox state of the water column and water-sediment interface back in time?

**Unfortunately, we do not have data on the redox state of the water column as it has never been undertaken. Mn is sometimes used as a proxy for redox conditions at the water-sediment interface (Jimenez-Espejo et al., 2019). Unfortunately, this element has not been measured in DTGC2011 core.**

Line 290: Again the concept that preservation of organic matter is better in surficial (younger) sediments is unfounded and in contradiction with section 3.4. It should be revised throughout the whole manuscript.

**We have amended the sentence (line 359)**

Lines 311-314 and thereafter: I don't think such a difference can be considered really significant (keeping in mind that the reproducibility was +/- 0.26 per mill). This statement might be a little far-fetched and I would suggest to remove it.

**We have amended the wording to make it clear this is a tentative suggestion only (lines 419).**

Lines 355-356. I am not convinced by this statement when looking at the d13C profile of the C24 FA which shows a clear trend towards lighter values (2-3 per mill) within the last 150 years. Authors are mentioning this trend later on (lines 375-376). Could this be linked to increased land plant inputs due to ice retreat?

**We have added a few additional sentences discussing the trend in the C24 d13C in relation to atmospheric CO2 d13C changes (lines 428-432).**

Lines 366-367: please temper with 'do not tend to produce high proportions of this compound'.

**We have amended the sentence as suggested (lines 443).**

Lines 372-378: This again is somewhat speculative. If both FA have distinct origins, than the diagenetic impact on their 13C composition may be significantly different. What about the possibility that either one or both FA are being produced in the top sediments?

**We disagree that the C24 and C18 would have such different diagenetic pathways. While we cannot rule out this suggestion completely, we are not aware of any literature to suggest this would be the case. As mentioned before, there is no evidence of any bioturbation or metazoan benthic activity in the sediments, and the bottom waters are known to be anoxic meaning any fatty acid producers within the sediments would be very limited.**

Section 4.3 (lines 522-530) and conclusions (lines 540-541): As considered for FA in actual figures 2 and 3, a correlation table would help in highlighting putative relationships between lipid biomarker (concentration or d13C) profiles and specific phytoplanktonic species.

**While we agree that this could be helpful, unfortunately the biomarker and diatom data were taken from adjacent samples (thus have different depths and ages) meaning they cannot be directly compared (unless they were resampled which would introduce errors). In contrast, comparison of different fatty acid compounds was possible since they which were all present within each sample meaning they were analysed simultaneously. Furthermore, due to the nature of the data, having high frequency-high amplitude changes due to dynamic environment, and the fact that they are different types of data, we do not think that a correlation table comparing diatoms and biomarkers would be useful. For this reason, it is not common in palaeoclimatic to look at correlations and is generally considered more useful to look at broad coherence between datasets which may change downcore. Therefore, we think it is more useful to plot the downcore records together in order to see the coherence between them.**

[revised manuscript text omitted]

Key:
- 0.9-0.99
- 0.8-0.89
- 0.7-0.79
- 0.6-0.69
- 0.5-0.59

**Figure 3**: $R^2$ values for fatty acid concentrations throughout core DTGC2011. Values are colour coded according to the key on the left. Black border denotes correlations within each group.

|  | C$_{16}$ | C$_{17}$ | C$_{18}$ | C$_{20}$ | C$_{22}$ | C$_{24}$ | C$_{26}$ |
|---|---|---|---|---|---|---|---|
| C$_{16}$ | | 0.74 | 0.87 | 0.80 | 0.24 | 0.09 | 0.21 |
| C$_{17}$ | 0.74 | | 0.73 | 0.72 | 0.28 | 0.08 | 0.19 |
| C$_{18}$ | 0.87 | 0.73 | | 0.93 | 0.21 | 0.07 | 0.20 |
| C$_{20}$ | 0.80 | 0.72 | 0.93 | | 0.39 | 0.15 | 0.31 |
| C$_{22}$ | 0.24 | 0.28 | 0.21 | 0.39 | | 0.46 | 0.68 |
| C$_{24}$ | 0.09 | 0.08 | 0.07 | 0.15 | 0.46 | | 0.42 |
| C$_{26}$ | 0.21 | 0.19 | 0.20 | 0.31 | 0.68 | 0.42 | |

Key:
- 0.8-0.99
- 0.6-0.79
- 0.4-0.59
- 0.2-0.39
- 0.0-0.19

**Figure 4**: $R^2$ values for fatty acid concentrations in core DTGC2011 below 25 cm only. Values are colour coded according to the key on the left. Black border denotes correlations within each group.

[Figure]

**Figure 35: Concentrations of the C18 fatty acid (blue), the HBI triene (red), HBI diene (grey) (Campagne, 2015), C24 fatty acid (orange) from core DTGC2011. The left-hand panels show 1550 to 1950 C.E. and the right hand panels show 1950 to 2000 C.E., plotted on different y-axes due to the elevated concentrations in the top part of the core. Grey vertical bands highlight coincident peaks in C18 fatty acid and HBI triene records.**

[Figure]

**Figure 64:** δ¹³C values of the C₂₄ fatty acid (orange) and relative abundances (%) of the open water diatom *Fragilariopsis kerguelensis* (green). Also shown are relative abundances of the four most abundant diatom groups in DTGC2011. *Chaetoceros* resting spores (CRS; grey line), *Fragilariopsis curta* group (dark blue line), *Fragilariopsis cylindrus* (purple line) and *Fragilariopsis rhombica* (light blue line). Thick line represents 3-point moving average for each. Grey vertical bands highlight periods where C₂₄ fatty acid δ¹³C is in phase with *F. kerguelensis*.

[Figure]

**Figure 57**: δ¹³C of the C₂₄ (orange) and C₁₈ (blue) fatty acid, HBI diene concentrations (green; plotted on a log scale) and relative abundances of *Fragilariopsis curta* plus *Fragilariopsis cylindrus* (purple). Latter two records reflect sea ice concentrations. Grey vertical band highlights period whe re low C₁₈ δ¹³C overlaps with elevated HBI diene concentrations.

---

## Referee Report (RR1)

[referee-annotated manuscript omitted]

---

## Author Response (AR2)

**Point-by-point response to comments**

Title: Fatty acid carbon isotopes: a new indicator of marine Antarctic paleoproductivity?
Journal: Organic Geochemistry

*Reviewer 1:*
*I was not one of the prior reviewers but it appears that the authors have thoroughly revised in response to the prior reviewers' comments.*

*While reservations remain about the generalist biomarkers and their inferences, the work is carefully reported and merits publication, as it usefully adds data to the conditions and questions posed.*

*A few minor issues are noted:*
*Line 127-53 – please attend to superscripts (13) and subscripts (FA and FAME) and italicization (n) in the track changes text this has been incompletely formatted, and elsewhere in the ms (line 359-361)*

**Response 1:** Amended as suggested.

*Line 160 – fully extract. The method was sonication, the HBIs are unlikely to be fully extracted. Please remove word 'fully'. The IS doesn't replicate analyte, but reporting its extraction efficiency would be appropriate. Query: As noted on lines 188 the standard was added after extraction for FAs, was it added before extraction for HBIs, i.e. were these separate extractions as stated or is this incorrectly stated? Please check and correct if needed.*

**Response 2:** Removed the word 'fully' as suggested. FAs and HBIs were extracted separately for different PhD projects. We have added a line at 151 to clarify this.

*Free-fatty acids only. As noted by a prior reviewer, bound FAs as well as free fatty acids would be pertinent to production and this is not satisfactorily resolved by simply stating the paper studies free FAs. I suggest adding a statement that the release of C16FAs is likely to be incomplete using the sonication approach, and there are references to cite on the likely extraction efficiency if this was not assessed here. Extraction of bound FAs by saponification would likely release more C16FA. Please include a brief indication of the limitations of the methodology, based on what can be found in the literature, such as might inform future studies.*

**Response 3**: We have modified from 'free' to 'freely extractable (using a standard solvent extraction protocol),' We feel our language in this regard is now appropriate. We are extracting the Fatty Acids that are freely extractable using standard solvent extraction protocols for paleoclimatology. We are fully aware that additional fatty acids can be released by more aggressive chemical treatments (or even further by using pyrolysis). We are also aware that more FAs could also be released from macromolecular material (kerogen, etc) that are not present in the cores. For these reasons, we believe this is not necessary to expand in this paper as it is not an organic geochemical method development paper and the audience understands this type of standard approach for Holocene paleoclimate.

*Conclusion 'producers…would require further elucidation' – as the prior reviewer comments*

*indicate this claim is overworked, it's unlikely to be resolvable, or more-importantly generalizable, for such a non-specific compound. Rephrase to not suggest the information instead that can be gleaned from general biomarkers in combination with evidence for producers.*

**Response 4**: We have amended the statement so that it now reads:

*"Although we have made parsimonious interpretations, there are clearly uncertainties in interpreting the FA $\delta13C$, and, as such various assumptions have been made. The primary producers of the $C_{18}$ and especially the $C_{24}$ FAs are a key source of uncertainty. Because these are general biomarkers, produced by many organisms, it is impossible to constrain entirely to one producer class. But with further work in the region, it could be possible further elucidate the most likely contributors. The possibility of inputs of FAs from multiple sources, in particular from organisms further up the food chain, has consequences for their interpretation since this could mean the $\delta13C$ FA is not fully reflecting just surface water conditions".*

Such a statement fully acknowledges the uncertainties involved in our study and suggests that further studies like ours will help to better decipher the signal carried by the FAs.

**Suggestions for revision or reasons for rejection (will be published if the paper is accepted for final publication)**

*The paper by Ashely et al presents a fairly high resolution record of fatty acid concentrations and delta 13C values of the most abundant ones (C18FA and C24FA), from offshore Antarctica, in an attempt to find out if this can be used to reconstruct changes in productivity. This productivity is important as a carbon sink, via the biological pump. Overall, the data are fine and should be published. The discussion also brings up valid arguments but appears a little limited, as are the presented data.*
*I have given more detailed comments in an annotated pdf, but here some main concerns:*
*1. It would be useful to also show the TOC profile, as well as bulk 13C, to get a sense of the carbon burial rate, and to place the FA results in a broader sedimentological framework. Expression of the FA concentration on a TOC basis would be useful.*

**Response 5:** Unfortunately, the TOC was not analysed and the data is not available.

*2. Have the authors considered measuring sterol concentrations, and sterol 13C contents, instead of FAs? Because sterols are clearly of phytoplankton original while FAs can also have a bacterial source. Please comment (we chose for FA, and not sterols, because...*

**Response 6:** We thank the reviewer for their suggestion and agree this would be a good idea for a line of enquiry in theory. The first author studied this core and a Holocene IODP core from the Adelie drift for her PhD. She explored sterol concentrations and distributions. Sterols were abundant but highly coeluting. Because of this we found that suitable separation and isolation of sterol compounds for compound specific isotope analyses was not possible within the time-frame of the PhD.

*3. A potential bacterial source of especially C18FA should be discussed (this would be an argument against FAs..)*

**Response 7:** We thank the reviewer for this suggestion. We have added several additional references to Table S2 in the Supplementary which summarizes key references on potential sources of fatty acids, including both bacteria and additional references on marine phytoplankton sources:

*Allen, E. E. & Bartlett, D. H. (2002) Structure and regulation of the omega-3 polyunsaturated fatty acid synthase genes from the deep-sea bacterium Photobacterium profundum strain SS9The GenBank accession numbers for the sequences reported in this paper are AF409100 and AF467805. Microbiology 148, pp. 1903-1913.*

*Allen, E. E., Facciotti, D. & Bartlett, D. H. (1999), Monounsaturated but Not Polyunsaturated Fatty Acids Are Required for Growth of the Deep-Sea Bacterium Photobacterium profundum SS9 at High Pressure and Low Temperature. Applied and Environmental Microbiology 65, 1710-1720, doi:10.1128/aem.65.4.1710-1720.1999.*

*Jónasdóttir, S. H. (2019) Fatty Acid Profiles and Production in Marine Phytoplankton. Marine Drugs 17, 151.*

We have noted potential contributions of bacterial FAs to the main manuscript at lines 224-225, 265-266.

*4. A discussion about the apparently low concentration of unsaturated FAs is in order.*

**Response 8:** We do discuss this over two paragraphs, from line 242 to 265. Specifically:

*"Many studies have shown that significant degradation of FAs occurs both within the water column and surface sediments as a result of microbial activity, and that there is preferential break down of both short-chained and unsaturated FA, compared to longer-chained and saturated FA (Haddad et al., 1992; Matsuda, 1978; Colombo et al., 1997)…*
*…. The complete lack of both unsaturated and short chained (fewer than 16 carbon atoms) FA compounds identified within DTGC2011 samples, even within the top layers, suggests that selective breakdown of compounds has already occurred within the water column and on the sea floor (before burial). Wakeham et al. (1984) assessed the loss of FAs with distance during their transport through the water column at a site in the equatorial Atlantic Ocean and estimated that only 0.4 to 2% of total FAs produced in the euphotic zone reached a depth of 389 m, and even less reaching more than 1,000 m depth, the vast majority of material being recycled in the upper water column. Their results also show a significant preference for degradation of both unsaturated and short chained compounds over saturated and longer chain length compounds. Although no studies into the fate of lipids within the water column exist for the Adélie region, the >1,000 m water depth at the core site would provide significant opportunity for these compounds to be broken down during transportation through the water column. It is likely, therefore, that the distribution of compounds preserved within the sediments will not be a direct reflection of production in the surface waters, and explains the preference for saturated FAs with carbon chain lengths of 16 and more."*

So in the discussion above we do clearly discuss this issue. We don't really know how much more of a discussion is in order for compounds that we could not measure.

*5. I find the apparent correlation between the C24 FA data and diatom species abundance not convincing; this needs to be made more clear.*

**Response 9:** We discuss correlations between the SCFAs ($R^2$ values between 0.97 and 0.99), and LCFAs (0.88 and 0.95) and summarize this in Fig. 3. These correlations are significant and the distinct groupings suggest that compounds within each group (SCFAs and LCFAs) likely have a common precursor organism or group of organisms.

However, with the relationship of the $C_{24}$ FA delta$^{13}$C data (presuming the reviewer is referring to the isotope data, they don't specify) we are careful not to talk of correlation. We feel that we do couch this part of our discussion in suitably cautious terms. For example we state at line 546: *"Comparison between $\delta^{13}C_{24FA}$ and the major diatom species abundances within the core (Fragilariopsis kerguelensis, Fragilariopsis curta, Fragilariopsis rhombica, Fragilariopsis cylindrus, Chaetoceros resting spores) shows a reasonably close coherence with Fragilariopsis kerguelensis, particularly since ~1800 C.E. (Fig. 6)."*

We are careful with our language and "Reasonably close coherence" is not the same as "correlation". We feel the readers can also judge visually through data comparison in Fig. 6. They are clearly some excursions which show coherence.

*6. Please include also the other FA 13C data, if not for the discussion in this particular paper then for future reference for others who might measure these on other locations. The C22FA appears to be about equally abundant as the C24FA (looking at the GC trace in the supplement); the same is valid for the C16FA (which is of course a very generic lipid).*

**Response 10:** Insufficient data of acceptable quality were measured on the $C_{16}$ FA and $C_{22}$ FA to be included in the study and manuscript.

---

## Author Response (AR3)

**Point-by-point response to comments**

Title: Fatty acid carbon isotopes: a new indicator of marine Antarctic paleoproductivity?
Journal: Organic Geochemistry

*This remains an interesting paper, revisions are minimal and improve the manuscript.*
*I was R1 on the last round. I am satisfied with revisions.*
We thank Prof. Feakins for the previous reviews and for her final suggestions which we have implemented (and detailed below). In addition we have added acknowledgements and author contributions.

*Minor formatting and technical corrections remain:*
*Line 52 – I read the rebuttal and the revision, 'freely extractable (using a standard solvent extraction protocol)' remains is obscure as yields differ between common solvent extraction methods (sonication, microwave, soxhlet and pressurized ASE or EDGE extraction). The method can be detailed in the methods. Here, suggest "free (solvent-extractable)" would suffice.*
Amended as suggested

*Line 110 – ultrasonication – please state the duration of sonication and any repetition of extraction cycles, with the goal of this to be repeatable by other scientists and to allow for comparability between sites and groups. Ideally specify the approx. mass of the samples, volume of solvent and number of extraction cycles and rinse steps, as these could affect extraction yield.*
Amended as suggested.

*Formatting carries over from the last review. Here are the specific items noticed on this reading:*
*Line 144, 120, 122, 124, 137, 138 – FAMEs not FAs (as analyzed as FAMEs on the GC).*
*Line 122 – subscript 19.*
*Line 125, 126 superscript and subscripts as appropriate for d13C, 12C, 13C, CO2*
*Line 144- 149, did you want to subscript FA and FAME when using after d13C, as on line 576 (either is fine, just be consistent)?*
*The n for normal, should be italicized.*
*Line 351-353 – the CL are not subscript, as they are elsewhere.*
*Line numbers refer to track changes ver*sion.
We have implemented all of the above minor formatting corrections.